# Nature of 2D XY antiferromagnetism in a van der Waals monolayer

Cheol-Yeon Cheon [1,2], Volodymyr Multian [1,2], Kenji Watanabe [3], Takashi Taniguchi [4], Alberto F. Morpurgo [1,2] ✉ & Dmitry Lebedev [1,2] ✉

Two-dimensional antiferromagnetism has long attracted significant interest in many areas of condensed matter physics, but only recently has experimental exploration become feasible due to the isolation of van der Waals antiferromagnetic monolayers. Probing the magnetic phase diagram of these monolayers remains however challenging because established experimental techniques often lack the required sensitivity. Here, we investigate antiferromagnetism in atomically thin van der Waals magnet $NiPS_3$ using magnetotransport measurements in field-effect transistor devices. Temperature-dependent conductance and magnetoresistance data reveal a distinct magnetic behavior in monolayers as compared to thicker samples. While bilayer and multilayer $NiPS_3$ exhibit a single magnetic phase transition into a zig-zag antiferromagnetic state driven by uniaxial anisotropy, monolayer $NiPS_3$ undergoes two magnetic transitions, with a low-temperature phase governed by in-plane hexagonal magnetic anisotropy. The experimentally constructed phase diagram for monolayer $NiPS_3$ matches theoretical predictions from the six-state clock and 2D-XY models incorporating hexagonal anisotropy.

Two-dimensional (2D) antiferromagnets (AFMs) have garnered growing interest because of their unique properties and the role they play in the context of complex physical phenomena, such as high-$T_C$ superconductivity and spin liquid states[1–5]. Until recently, experiments on 2D AFMs focused on layered bulk crystals and relied on the assumption of weak magnetic coupling between adjacent crystalline planes[6,7]. However, the rapid advances in the synthesis and isolation of 2D van der Waals (vdW) magnets down to monolayer thickness have changed the situation, enabling the investigation of truly 2D AFMs[8,9]. The broad scope of available vdW magnets allows magnetism to be studied for a wide range of parameters, such as exchange interaction, anisotropy, and spin dimensionality, which result in different magnetic states[10–12]. In addition, the unique nature of vdW materials allows these parameters to be tuned via external experimental techniques not applicable to bulk systems, for instance gating and heterostructure engineering (including twisted moiré layers)[13,14].

Among various vdW AFMs, materials composed of ferromagnetically ordered monolayers with alternating magnetization between adjacent layers has so far attracted the most attention[14–16]. This is primarily because the magnetization of individual layers and few-layer stacks can be directly detected through experimental techniques such as scanning magnetometry and magneto-optical Kerr effect measurements[15]. In contrast, much less progress has been made in studying intralayer AFMs, where each individual monolayer exhibits antiferromagnetic ordering. Due to the absence of a net magnetic moment, experimental methods commonly employed to probe antiferromagnetism lack the sensitivity required to detect and characterize true AFM monolayers in these systems[9].

To address this issue, we focus on the vdW intralayer antiferromagnet $NiPS_3$[17–20]. There is broad consensus that below the Néel temperature $T_N$ of approximately 155 K, bulk crystals enter a long-range-ordered zig-zag AFM state[17]. However the nature of magnetism in few-

[1]Department of Quantum Matter Physics, University of Geneva, Geneva, Switzerland. [2]Department of Applied Physics, University of Geneva, Geneva, Switzerland. [3]Research Center for Electronic and Optical Materials, National Institute for Materials Science, 1-1 Namiki, Tsukuba, Japan. [4]Research Center for Materials Nanoarchitectonics, National Institute for Materials Science, 1-1 Namiki, Tsukuba, Japan. ✉e-mail: alberto.morpurgo@unige.ch; dmitry.lebedev@unige.ch

layer and monolayer NiPS$_3$ remains unresolved, because studies based on various Raman and magneto-optical techniques have not given a conclusive answer[18,21–24]. In particular, the spin-flop metamagnetic transition in NiPS$_3$–a hallmark of long-range AFM order–has so far only been observed in bulk and not in few-layer or monolayer crystals.

Here, we employ electrical transport measurements as a function of temperature ($T$) and magnetic field ($B$) to probe magnetic order and the associated spin-flop transition in NiPS$_3$ down to monolayer thickness. We find that thick multilayers exhibit a behavior consistent with the known magnetic state of bulk NiPS$_3$, which persists virtually unchanged down to bilayer thickness. In contrast, NiPS$_3$ monolayer first undergoes a transition into a phase which lacks long-range order before entering the low-temperature long-range ordered state governed by hexagonal magnetic anisotropy, and not by uniaxial anisotropy as for thicker crystals. These observations establish previously undetected magnetic phases of monolayer NiPS$_3$ and highlight the critical role of interlayer coupling in determining the magnetic ground state of ultrathin magnets.

## Results

### NiPS$_3$ field-effect transistors

Temperature-dependent magnetotransport measurements in tunnel junctions[8,25–27] or in field-effect transistor (FET) devices[28–30] were shown to be powerful for studies of ultrathin vdW magnets, enabling construction of their magnetic phase diagrams as a function of thickness. We adopt this strategy for NiPS$_3$ and measure transistors realized with multilayers of different thickness, all the way down to the ultimate monolayer. In the bulk, NiPS$_3$ has readily accessible $T_N \approx 155$ K, below which an intralayer zig-zag collinear AFM order is established, with spins predominantly lying in the **a-b** plane (Fig. 1a)[17]. Within this plane, the system is characterized by a uniaxial magnetic anisotropy, resulting in a sharp spin-flop transition just above $B = 10$ T[31]. We use this established knowledge to identify features in temperature-dependent magnetoresistance ($MR$) measurements and associate them with magnetic phase transitions.

FET devices were realized using NiPS$_3$ multilayers of different thickness (~13 L, 6 L, 2 L, and 1 L), which are fully encapsulated between h-BN flakes, and employ few-layer graphene strips as source/drain contacts. Contrary to previous reports, in which source/drain contacts were defined by electron-beam lithography and metal deposition directly onto NiPS$_3$[32,33], graphene contacts result in a higher device quality, and allow proper transistor operation to be observed down to monolayer thickness, even at cryogenic temperatures. Irrespective of thickness, all our devices exhibit n-type semiconducting behavior (Fig. 1b) with electron mobility values of 0.5-3 cm$^2$V$^{-1}$s$^{-1}$ at 10 K estimated from the transistor transfer curves (see Supplementary Fig. S1). These values are comparable to those observed in the best transistors of other vdW magnetic semiconductors, such as CrPS$_4$[30], and NiI$_2$[29,34].

The high quality of our transistor devices enables the detection of the phase transitions by simply looking at the temperature dependence of the device conductance ($G$). Indeed, upon cooling the device, the conductance exhibits a kink at a specific temperature, which can be determined from the position of the peak in the derivative of the conductance $dG/dT$. For the thickest layers (13 L and 6 L NiPS$_3$) the position of the peak is around 150 K, matching closely the value of the transition from the paramagnetic state to the AFM zig-zag state of bulk NiPS$_3$ (Fig. 1c). We find that the $dG/dT$ peak position has lower values in devices with thinner NiPS$_3$ crystals, which is in line with previously reported values down to bilayer thickness[18].

For 1 L NiPS$_3$, earlier experimental work could not find convincing evidence of a magnetic transition[18]. The only claim for the occurrence of a low-temperature magnetic phase in monolayer NiPS$_3$ reported so far relies on the analysis of a very broad background in Raman signal, attributed to two-magnon scattering[22]. However, as the background is also present at room temperature, its relation to a magnetic phase can

be questioned and deserves future studies[18]. In contrast, our transistor measurements show that a peak in $dG/dT$ is clearly visible also in monolayer transistor devices (see Fig. 1c bottom panel), allowing us to confidently establish the occurrence of a transition. We infer a value of the critical temperature for 1 L NiPS$_3$ of 124 K, significantly lower than 140–150 K found for bilayers and thicker multilayers. The transition temperature in monolayer NiPS$_3$ was reproduced in two additional 1 L transistors; see Supplementary Fig. S2. Lastly, the position of the peak in $dG/dT$ does not depend on gate voltage or on other parameters of the transistor, as expected for a manifestation of a phase transition (see Supplementary Fig. S3).

### Magnetic phase diagram of multilayer NiPS$_3$

To explore the nature of the magnetic transitions in monolayer, it is essential to establish a measurement protocol that enables us to construct the phase diagram of NiPS$_3$ multilayers of different thicknesses. We begin by analyzing the 6 L device, whose magnetic response, as we show below, is in full agreement with the known magnetic properties of bulk NiPS$_3$. We recall that the magnetic structure of bulk NiPS$_3$ is governed by a strong easy-plane anisotropy ($D^z \approx 0.2$ meV per Ni), which confines the spins to predominantly lie in the **a-b** plane, and a weak uniaxial anisotropy ($D^x \approx -0.01$ meV per Ni)[35,36]. The latter orients the spins (and hence the Néel vector, **L**) along the crystallographic **a**-axis (Fig. 1a). Application of an external magnetic field along this axis results in a reorientation of a Néel vector at a critical field value $B_{sf}$–a spin-flop metamagnetic transition[31]. We expect that measuring the $MR$ with the field aligned to the Néel vector can probe this transition.

To reveal the orientation of the Néel vector of the 6 L NiPS$_3$ crystal in the FET device, we perform low temperature polarization-resolved photoluminescence (PL) measurements. Consistent with the previous reports[37], we find that the 6 L sample shows a sharp PL peak at $E = 1.475$ eV (Fig. 2a) with a high degree (80%) of linear polarization (Fig. 2b). The direction of linear polarization corresponds to the direction of **L**[31,38], which is oriented along one of the sample edges (see the inset of Fig. 2a). By checking multiple spots, we confirm a uniform orientation of **L** in the entire device channel (Supplementary Fig. S4).

Measurements of the 6 L FET device with magnetic field applied along the Néel vector exhibit a negative magnetoresistance $MR$ ($MR(B) = \frac{R(B)-R(0)}{R(0)} \times 100\%$) that emerges below $T_N = 149$ K (as determined from the peak in $dG/dT$), confirming the onset of magnetic order (Fig. 2c). At lower temperature, we observe an abrupt decrease in resistance above 10 T, which we assign to the spin-flop transition (Supplementary Fig. S5)[31,38]. The magnitude of the $MR$ at $B=B_{sf}$ increases upon cooling, reaching a value of 0.9% at $T = 5$ K, which is highly reproducible over repeated cool-down–warm-up cycles (Supplementary Fig. S6). This observation underscores the robustness of both the Néel vector orientation and our measurement protocol.

The 2D plot of $MR$ as a function of magnetic field and temperature maps the magnetic phase diagram of 6 L NiPS$_3$, displaying the paramagnetic, collinear AFM, and spin-flop phases (Fig. 2d). The critical spin-flop field increases with temperature, analogously to other weakly anisotropic AFMs such as MnPS$_3$[8], MnF$_2$[39], and CuMnAs[40]. Its temperature dependence is consistent with the expression $H_{sf}^2 = \frac{2K}{\chi_\perp - \chi_\parallel}$ calculated within a molecular-field approximation[41], where $K$ is the uniaxial anisotropy constant (per unit of volume) and $\chi_\parallel$ ($\chi_\perp$) is the in-plane magnetic susceptibility parallel (perpendicular) to the Néel vector. Comparing previously reported magnetic susceptibility data for bulk NiPS$_3$[17] with our $B_{sf}$ vs $T$ measurements of the 6 L device (Fig. 2e) enables us to estimate an anisotropy energy of $D^x \approx -0.003$ meV (per Ni atom), close to the value of $-0.01$ meV obtained from neutron scattering[35,36]. This correspondence demonstrates the capability of magnetotransport for studying the temperature evolution of intralayer AFM order at a quantitative level.

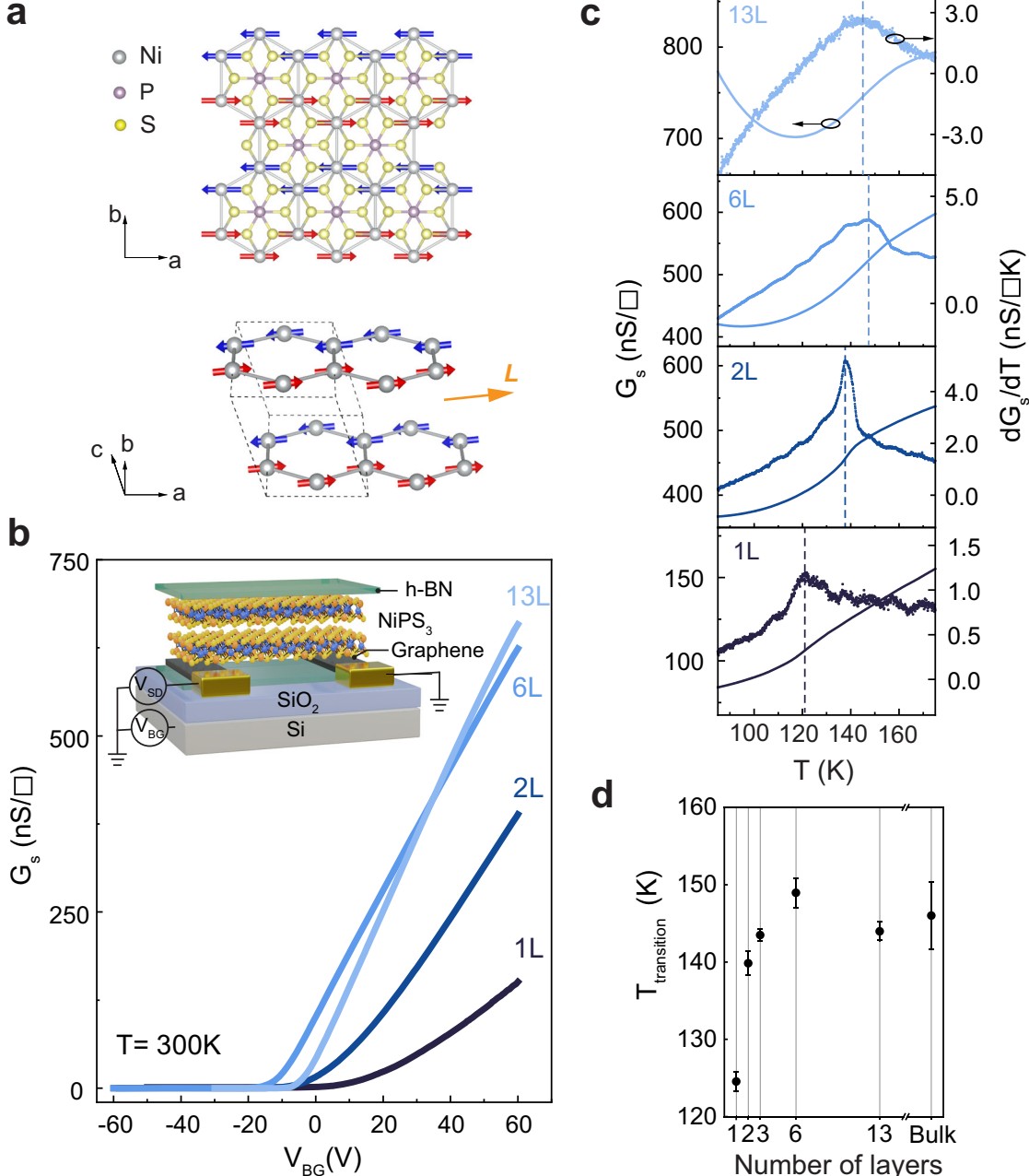

**Fig. 1 | Electrical transport in NiPS₃ down to monolayer thickness. a** Top view of monolayer NiPS₃ (top panel) and stacking of layers in NiPS₃ structure (bottom panel, only Ni atoms are shown, dashed line is a unit cell). Red and blue arrows mark Ni spins of two magnetic sublattices and define the Néel vector **L**. **b** Room temperature sheet conductance ($G_s$) of 13 L, 6 L, 2 L, and 1 L NiPS₃ FET devices as a function of back-gate voltage ($V_{BG}$). The inset shows the schematic of a 2 L NiPS₃ FET device and the measurement configuration. **c** Variable temperature sheet conductance (left axis) and its derivative ($dG_s/dT$, right axis) of 13 L, 6 L, 2 L, and 1 L NiPS₃. The peak of the $dG_s/dT$ corresponds to magnetic phase transition, marked by vertical dashed lines. Measurements were performed with the following $V_{SD}/V_{BG}$ configurations: 2 V/90 V (13 L), 4 V/80 V (6 L), 2 V/100 V (2 L), and 2 V/80 V (1 L). **d** Transition temperature as a function of layer number. Error bars are the standard deviation of the transition temperatures collected from the measurements with different $V_{BG}$, and/or from additional samples.

The results just discussed show that *MR* measurements on transistor devices allow the detection of the spin-flop transition in few-layer NiPS₃—which so far had been reported only in NiPS₃ bulk crystals—and enable mapping of their magnetic phase diagram as a function of temperature and field. The experiments appear to be in full agreement at a quantitative level with the same long-range zig-zag AFM state present in bulk NiPS₃ crystals, as indicated by the value of the spin-flop field. This conclusion is worth emphasizing, because it has been recently proposed that NiPS₃ layers with a thickness below 10 nm (approximately 15 L) enter a so-called nematic vestigial state with no

long-range magnetic order, which differs from the magnetic state of bulk NiPS₃ crystals[24].

## Magnetic phase diagram of 1 L NiPS₃

Having established a protocol to probe the magnetic phase diagram of NiPS₃ multilayers, we investigate whether a change in the behavior of the measured *MR* and hence the magnetic state is observed below some critical thickness. We start by comparing low-temperature *MR* and corresponding spin-flop transitions of 2 L and 1 L NiPS₃ FETs (see Fig. 3a, b). Since the suppressed PL in 1 L and 2 L NiPS₃ makes the

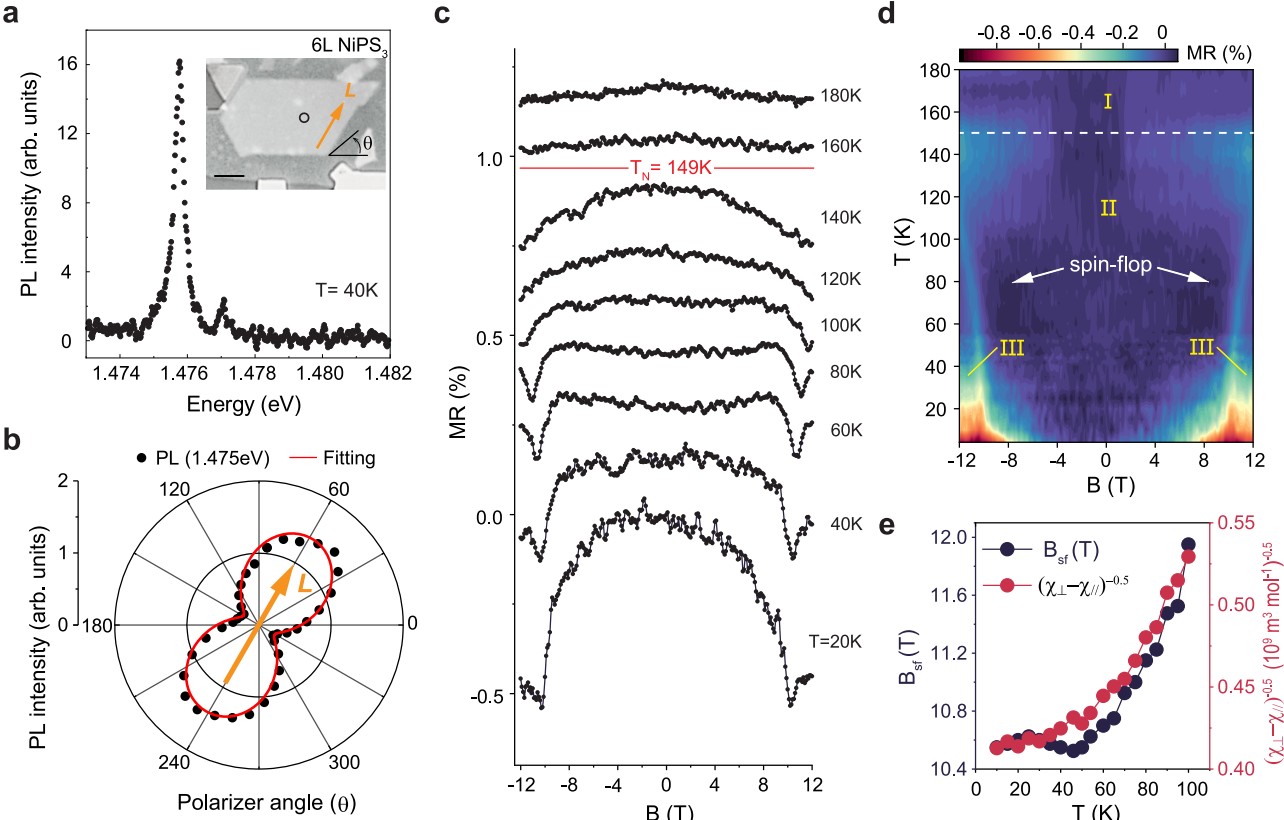

**Fig. 2 | Spin-flop transition in multilayer NiPS₃ probed via in-plane *MR*. a** Low-temperature photoluminescence (PL) spectrum of 6 L NiPS₃ displaying a sharp emission at $E = 1.476$ eV. The inset shows the device image with the optical excitation area (black empty circle). The scale bar is 4 μm. **b** PL intensity as a function of the angular position ($\theta$) of the detector linear polarizer. PL is linearly polarized along the crystal edge (see inset in **a**), which is the orientation of its **L**. **c** In-plane *MR* with **B**//**L** as a function of temperature measured at $V_{SD} = 4$ V and $V_{BG} = 80$ V. The data are shown with vertical offsets for clarity. **d** Magnetic phase diagram of 6 L NiPS₃ (I: paramagnetic phase, II: collinear AFM phase, III: spin-flop phase). $T_N = 149$ K (white dash line) remains the same with in-plane magnetic field strength up to 12 T. **e** Comparison between the estimated spin-flop field and the inverse square root of in-plane magnetic susceptibility anisotropy of bulk NiPS₃ as a function of temperature. The susceptibility data are taken from a reference[17].

identification of the Néel vector direction by optical means practically impossible[42], *MR* measurements were taken with the field along three distinct crystal edges 120° apart from one another to align the field with all possible Néel vector orientations (Fig. 3a). The feasibility of this approach is supported by the strong tendency of the exfoliated crystals to have zig-zag termination[43], as demonstrated in thicker 6 L and 13 L samples (for the results of 13 L; see Supplementary Fig. S7).

The 2 L NiPS₃ device exhibits an overall behavior that is virtually identical to that observed in thicker NiPS₃ multilayer devices. At 40 K, the *MR* shows a sharp change at approximately $B = 10$ T when the field is oriented along sample edge 1 (Fig. 3b), which we assign to the orientation of its Néel vector. Similarly to the 6 L sample, we observe that the spin-flop field in the 2 L device increases at higher temperature (see Supplementary Fig. S8). When we apply the magnetic field along the other two zig-zag edges, we observe smooth *MR* curves without abrupt transitions, consistently with the presence of uniaxial anisotropy, as for the 6 L sample and bulk NiPS₃[31].

The *MR* of the monolayer device is drastically different. First, at 40 K the spin-flop transition occurs at a critical field of approximately $B = 5$ T, rather than 10 T as in bilayers and thicker NiPS₃ multilayers (Fig. 3d). This indicates smaller magnitude of in-plane anisotropy of 1 L NiPS₃ compared to multilayer samples. Second, the transition is observed when the magnetic field is aligned parallel to any of the three edges (Fig. 3c), with a sharp *MR* decrease observed in all cases, and with a *MR* value and functional dependence that is the same irrespective of the edge along which the magnetic field is aligned. We have also measured MR with the field along the intermediate direction

(perpendicular to the edge 1 on Fig. 3c) where the shape of MR differs from that observed with fields along the edges 1-3 (Supplementary Fig. S9). These observations unambiguously demonstrate that that the magnetic state of 1 L NiPS₃ is governed by hexagonal magnetic anisotropy, and not uniaxial as for multilayer crystals.

The unique behavior of 1 L NiPS₃ becomes even more apparent when looking at the full temperature dependence of the *MR* measured with **B** applied along one of the crystal edges (Fig. 4a). A first important difference with respect to thicker multilayers is that in monolayer devices no *MR* is observed when the temperature is lowered below the critical temperature $T_C$ identified by the peak in $dG/dT$ ($T_C = 124$ K). We find that *MR* appears only below $T^* = 60$ K, past which it increases continuously upon cooling (see Fig. 4b). Also, it is only for $T < T^*$ that the spin-flop transitions in 1 L NiPS₃ devices become visible in the measured *MR*. Finding a temperature $T^*$ much lower than $T_C$ below which *MR* and the spin-flop transition appear indicates the existence of two magnetic transitions in 1 L NiPS₃. These observations highlight the unique behavior of monolayers as compared to 2 L and thicker NiPS₃ multilayers. By measuring the complete set of *MR* curves at different temperatures and *G* vs *T* curves, we construct the magnetic phase diagram of 1 L NiPS₃, which is shown as a color plot of Fig. 4c.

Finally, in the vicinity of $T^*$, we observe noticeable current fluctuations during the magnetic field sweeps (Fig. 4d), a behavior not observed in any of the thicker NiPS₃ multilayers. The fluctuations, which appear as spontaneous random switching between two resistance levels, are also observed in the absence of a magnetic field (see Supplementary Fig. S10). The corresponding histogram of resistance

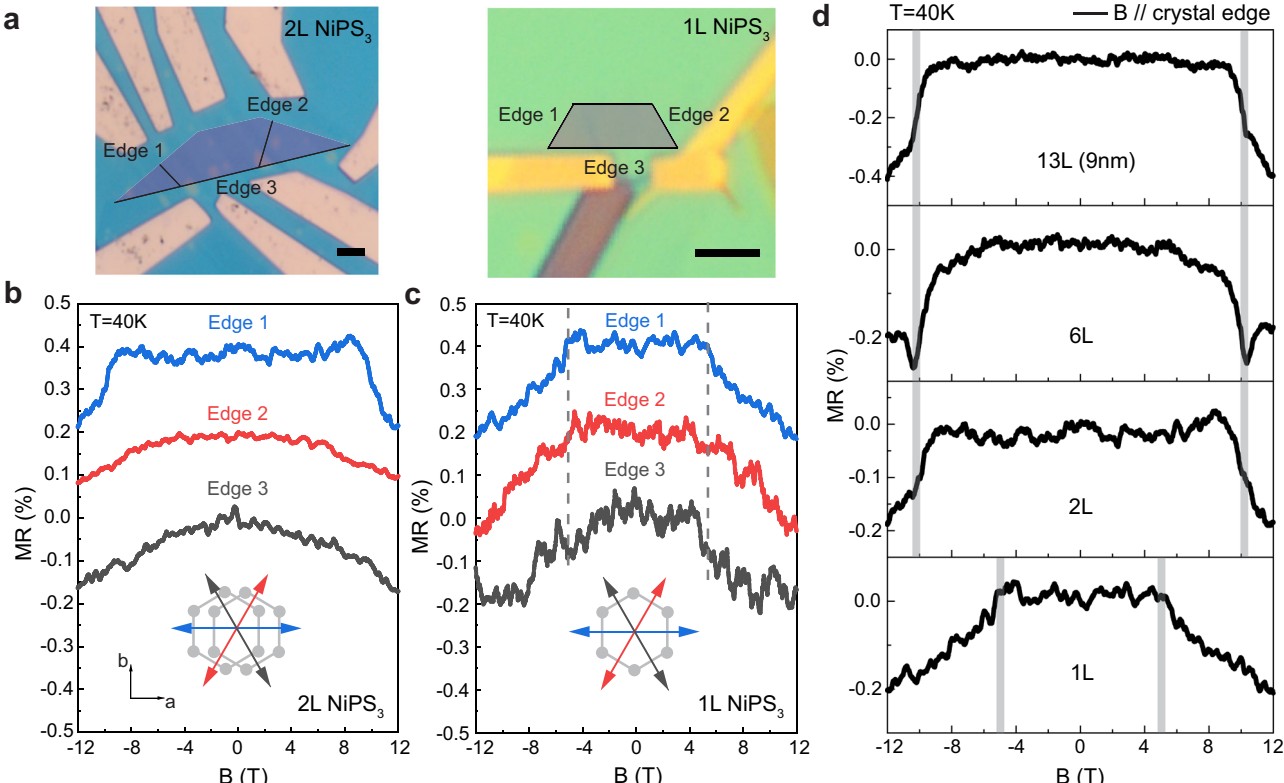

**Fig. 3 | Spin-flop transition in 2 L and 1 L NiPS₃. a** Optical images of 2 L and 1 L NiPS₃ FET devices. NiPS₃ flakes are highlighted in color and their 120° crystal edges are labeled for reference. Scale bars are 3 µm. *MR* of 2 L (**b**) and 1 L (**c**) at *T* = 40 K with the magnetic field applied along the three crystal edges. Measurements were taken at $V_{SD}$ = 4 V and $V_{BG}$ = 80 V (2 L) and $V_{SD}$ = 2 V and $V_{BG}$ = 100 V (1 L). Data are shown with vertical offsets for clarity. The critical fields for 1 L are marked by gray dashed lines. Insets in (**b**, **c**) show top views of Ni honeycomb lattices for 2 L and 1 L NiPS₃ with arrows indicating the applied magnetic fields along the three zig-zag orientations. **d** *MR* of 13 L, 6 L, 2 L, and 1 L NiPS₃ with **B** along **L** (or edge 1) at *T* = 40 K. The critical spin-flop fields shown as vertical gray lines remain unchanged down to 2 L, and are notably lower for 1 L NiPS₃. Measurements were taken at following $V_{SD}/V_{BG}$ configurations: 1.5 V/80 V (13 L), 4 V/80 V (6 L), 4 V/80 V (2 L), and 2 V/100 V (1 L).

values shows a bimodal distribution only at *T* = *T*\* = 60 K (see right column of Fig. 4d). While the exact origin of the fluctuations and their coupling to magnetic order in 1 L NiPS₃ is currently unclear, we note that pronounced fluctuations attributed to the critical spin dynamics near a ferromagnetic transition have been recently reported in monolayer CrBr₃[44] near its Curie temperature. For NiPS₃, finding that these fluctuations are only present in monolayers and only when *T* = *T*\* provides additional evidence for the occurrence of a magnetic transition at *T*\*.

## Discussion

The magnetic phase diagrams of NiPS₃ multilayers of different thickness reconstructed from temperature-dependent *MR* measurements highlight the unique behavior of 1 L NiPS₃. For 2 L and thicker samples, we observe an AFM phase at zero magnetic field below 140–150 K and a low-temperature transition to the spin-flop phase occurring just above 10 T, indicative of the presence of an uniaxial magnetic anisotropy akin to that of bulk NiPS₃. As mentioned earlier, recent Raman and nitrogen-vacancy diamond spin relaxometry studies of NiPS₃ multilayers with thickness below 10 nm (from 2 L to approximately 15 L)[24] have been interpreted in terms of a vestigial nematic phase, in which the multilayers lack long-range magnetic order and only feature short-range spin correlations. In contrast to this conclusion, our magnetoresistance data for 2 L, 6 L, and 13 L NiPS₃ provide no indications that the nature of AFM order in bilayers and thicker multilayers deviates from that of bulk crystals[31,45]. A final assessment of which state better explains the existing experimental data–a bulk-like long-ranged zig-zag AFM order or the newly

proposed vestigial nematic phase–will require the investigation of how the vestigial nematic phase responds to an external magnetic field, which is currently unknown.

Possibly the most intriguing conclusion of our experiments is that, in contrast to multilayers, 1 L NiPS₃ is found to undergo two distinct magnetic phase transitions: the first at $T_C$ = 124 K, revealed by a peak in the *dG/dT* vs *T* measurements, and the second transition at *T*\* = 60 K, revealed by the enhanced fluctuations of electrical current and the onset of *MR*. Only below *T*\*, 1 L NiPS₃ exhibits a spin-flop field close to 5 T, approximately half the value observed in 2 L and thicker multilayers. The *MR* measured below *T*\* also clearly shows that 1 L NiPS₃ has a hexagonal magnetic anisotropy, and not uniaxial anisotropy as in thicker multilayers, a behavior consistent with its higher symmetry (D₃d point group) as compared to monoclinic C2/m symmetry for multilayers (C₂h point group). It is clear from these observations that the truly 2D limit of magnetism in NiPS₃ is only reached when the thickness is reduced to an individual monolayer. A key difference between monolayers and thicker multilayers comes from the symmetry of magnetic anisotropy (hexagonal in monolayers and uniaxial in thicker multilayers), strongly suggesting that the uniaxial anisotropy in bi- and thicker layers originates from interlayer interactions. These results make it therefore apparent that interlayer interactions drastically modify the magnetic phase diagram of NiPS₃, illustrating with a concrete example that treating bulk materials as 2D because of the weak coupling between adjacent crystalline layers can in general be extremely misleading.

The phase diagram derived from *MR* data of 1 L NiPS₃ is in full agreement with renormalization group calculations that incorporate

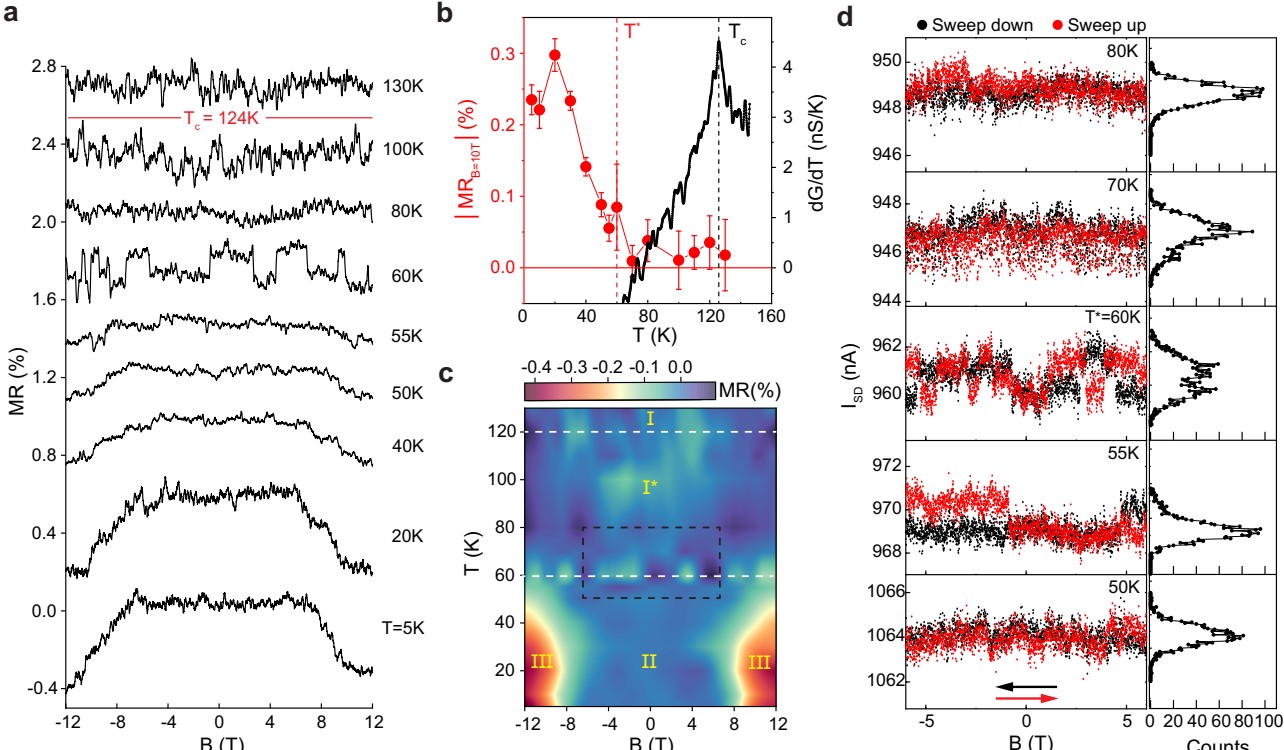

**Fig. 4 | Phase diagram of 1 L NiPS₃. a** *MR* of 1 L NiPS₃ as a function of temperature, measured at $V_{SD} = 2$ V and $V_{BG} = 100$ V. *MR* and spin-flop transition emerge below $T = 60$ K, at which current fluctuations also appear. **b** Absolute value of *MR*(%) at $B = 10$ T (left axis, red) and $dG/dT$ (right axis, black) as a function of temperature measured at $V_{SD} = 2$ V and $V_{BG} = 100$ V. The error bars in the *MR* data are calculated based on the standard deviation of the *MR* signal within the field range of ±3 T. $T_C$ (black dashed line) and $T^*$ (red dashed line) mark the temperatures where $dG/dT$ peaks and *MR* emerges, respectively. **c** Phase diagram of 1 L NiPS₃ (I: paramagnetic phase, II: long-range ordered AFM phase, III: spin-flop phase, and I*: Quasi-long-range ordered phase). The horizontal white dashed lines mark $T_C$ and $T^*$, respectively. **d** Fluctuations of electrical current observed during magnetic field sweeping in the vicinity of $T^*$ (left panel), within the magnetic and temperature range indicated by the black dashed rectangle in (**c**). Right panel shows corresponding frequency counts (from down sweeps).

crystal-field perturbations[46]. These perturbations enable a detailed analysis of how different magnetic anisotropy terms influence the magnetic state within the framework of the 2D XY model[16,46]. According to the calculations by José, J.V. et al.[46], a 2D XY system subject to weak hexagonal in-plane anisotropy is expected to undergo two phase transitions, whereas for uniaxial anisotropy only one transition into a long-range AFM state is predicted to occur. The system subjected to hexagonal anisotropy first undergoes a Berezinskii–Kosterlitz–Thouless (BKT) transition to a state without long-range order[47,48] at a higher temperature $T = T_{BKT}$. At a lower temperature the system undergoes a second transition due to spontaneous symmetry breaking, establishing long-range order with the Néel vector along one of six symmetry-equivalent degenerate directions[19]. A similar description can be also provided by the so-called *p*-state clock model, in which the spins point only along discrete in-plane angles ($\theta = 2\pi n/p$, with $n = 1, 2, ..., p$). Studies of such systems show that for $2 \le p \le 5$ only one phase transition into an ordered phase occurs, whereas for $5 \le p \le \infty$ two phase transitions are expected, with an intermediate XY-like phase between the low-temperature ordered and the high temperature disordered phases[49–51].

These theoretical scenarios naturally explain our results for monolayers if we identify $T_C$ with $T_{BKT}$ and $T^*$ with the transition temperature into a long-range AFM state. The observed absence of *MR* and spin-flop transitions in the temperature range between $T^* = 60$ K and $T_C = 124$ K is consistent with the absence of long-range order characteristic of the BKT phase transition. The emergence of *MR* and spin-flop transitions when the field is aligned along any of the crystalline zig-zag directions signals that the system enters a phase with long-range AFM order. Such phase is highly likely to have a multi-

domain structure, where different regions of the monolayer have Néel vector pointing along different zig-zag directions (this is also likely the reason why conventional Raman measurements fail to detect magnetism in 1 L NiPS₃[21]). Finally, theory also explains why bi- and thicker layers exhibit only one magnetic transition into a long-range ordered AFM state, as the behavior of NiPS₃ crystals thicker than 1 L is governed by uniaxial anisotropy.

## Methods

### Device fabrication

NiPS₃ crystals were obtained by CVT growth (HQ Graphene) and handled in a nitrogen-filled glovebox under sub-ppm oxygen and water levels. The crystals of few-layer NiPS₃, h-BN, and graphite on a SiO₂/Si substrate were obtained by mechanical exfoliation. The thickness of NiPS₃ was identified based on the optical contrast (see Supplementary Fig. S11) and atomic force microscopy measurements. The heterostructures were fabricated using the vdW crystal pick-up transfer method using polycarbonate (PC) film inside the glovebox[52]. Electron-beam lithography was used to pattern PMMA polymer masks. Reactive ion etching (with CF₄) was performed to etch h-BN to make the edge contacts to the few-layer graphite[53]. The contact electrodes were made by electron-beam evaporation of Cr 20 nm/Au 60 nm.

### Low temperature magnetotransport measurement

Transport measurements were performed in an Oxford cryostat equipped with a superconducting magnet. Gate and bias voltages were applied using Keithley 2400 sourcemeter and homemade low-noise source meter, respectively. The current was measured using a homemade low-noise amplifier and an Agilent 34410 A multimeter. The

relative alignment of the crystal edge with respect to the direction of magnetic field was performed using an optical microscope when mounting the device at room temperature. We estimate the alignment error in our measurements to be less than 5°.

**Low temperature photoluminescence measurement**

PL spectra were recorded using a Horiba LabRAM HR Evolution spectrometer with a grating of $1800\,gr\,mm^{-1}$ that provides a resolution of $0.01\,nm$. For polarization-resolved measurements, a continuous wave laser with a wavelength of $532\,nm$ was used. The polarization state of light emitted by the laser was converted into circular right polarization with the use of a quarter wave plate (ThorLabs AQWP05M-600). The laser light was focused on a ~$0.6\,\mu m$ (FWHM) spot using a window-corrected 63× objective. Si substrate with the device was glued on the copper plate of a He flow cryostat (Konti Micro from CryoVac GMBH) using GE Low Temperature Varnish. The power of the laser was adjusted to be $200\,\mu W$ at the output of the objective in order to achieve a reasonable signal-to-noise ratio with minimal effect of laser heating. The measurements were performed at the sample temperature of approximately $40\,K$, which was verified by fitting Stokes and anti-Stokes ratio of the Raman spectrum[54]. Polarization resolved measurements were performed by rotation of the analyzer (EdmundOptics, Dichroic Polarizing Film on Glass, spectral range 400–700 nm, extinction ratio 100:1) mounted in a custom-made motorized rotation stage at the entrance of the spectrometer. To minimize the distortion of polarization inside the spectrometer a depolarizer (ThorLabs DPP25-A) was placed in between the analyzer and the entrance of the spectrometer. In addition, a polarization-resolved sensitivity calibration was performed by measurement of light from a calibrated halogen lamp placed in front of the objective.

## Data availability

The data that support the findings of this study are available in the Yareta repository of the University of Geneva (https://doi.org/10.26037/yareta:5otraeqhnfaczja6dbxgjv5poa).

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

## Acknowledgements

The authors gratefully acknowledge Alexandre Ferreira for technical support. The authors thank Prof. Nicolas Ubrig for the valuable discussions. D.L. acknowledges the Swiss National Science Foundation (project PZOOP2_208760). A.F.M. gratefully acknowledges financial support from Division II of the Swiss National Science Foundation, under project 200021_219424. K.W. and T.T. acknowledge support from the JSPS KAKENHI (Grant Numbers 20H00354 and 23H02052) and World Premier International Research Center Initiative (WPI), MEXT, Japan.

## Author contributions

D.L. initiated the project. C.Y. fabricated the devices and performed the transport measurements with the help of D.L. V.M. performed optical measurements with the help of C.Y. K.W. and T.T. grew the h-BN crystals. D.L. and A.F.M. supervised the project. C.Y., D.L., and A.F.M. analysed the data. C.Y., D.L., and A.F.M. wrote the manuscript with the input from all the authors.

## Competing interests

The authors declare no competing interests.
