## [Transparent Peer Review file · Nature Communications]

Nature of 2D XY antiferromagnetism in a van der Waals monolayer

Corresponding Author: Dr Dmitry Lebedev

Version 1:

Reviewer comments:

Reviewer #1

(Remarks to the Author)

Please find the attachment: review_nat_comm_NiPS3.pdf

Reviewer #2

(Remarks to the Author)

Cheon et al. present findings from electrical transport measurements conducted on few-layer NiPS₃ FET devices. They identify a long-range antiferromagnetic (AFM) state in bi- and thicker-layer NiPS₃ by observing the spin-flop transition through the characterization of magnetoresistance in relation to the in-plane magnetic field. In contrast, they do not detect spin-flop-induced resistance changes in monolayer devices, leading them to assert that NiPS₃ represents the first experimental realization of a 2D XY antiferromagnet experiencing the Berezinskii-Kosterlitz-Thouless transition. While the experimental results are intriguing, questions emerge regarding whether these findings adequately support the authors' conclusion and the novelty of their work for publication in Nature Communications.

1. The authors claim that monolayer NiPS₃ undergoes a BKT transition, based on the observation that no abrupt change in magnetoresistance is observed above 10 T. However, it is important to note that there are several other possible explanations for this experimental observation, which the authors do not explicitly rule out. For instance, recent studies have shown that NiPS₃ transitions from a three-dimensional antiferromagnetic state to a two-dimensional "vestigial nematic" state due to strong quantum fluctuations as its thickness decreases. In this scenario, the spins are randomly canted away from the zigzag chain direction, leading to a loss of long-distance spin coherence. The authors must present additional evidence to establish a robust connection between their experimental findings and the proposed conclusion.

2. Several groups have shown that the BKT transition may occur in monolayer NiPS₃ (e.g., 10.1103/PhysRevB.107.L220407) using experimental techniques like helicity-resolved Raman and ultrafast spectroscopy. In this paper, the authors employ a different approach (electrical transport) to reach a similar conclusion, which diminishes its novelty and significance.

3. The manuscript is titled "Nature of 2D XY Antiferromagnetism in a Van der Waals Monolayer." However, much of the main text centers on the experimental results from a 6-layer device. A different title that more accurately reflects the content of the manuscript might be more appropriate.

4. According to the text, conducting polarization-dependent PL measurements on a monolayer device is challenging. How do the authors determine the crystal direction? Furthermore, the authors state that there are three types of magnetic domains; what are the sizes of these domains, and how are they distributed?

5. Typically, monolayer van der Waals materials are less stable than multilayer samples. Sample degradation could lead to changes in electrical and magnetic properties. Did the authors assess the stability of the device?

Due to these reasons, I cannot recommend publication of this work in Nature Communications. It may be suitable for other specialized journal after major revisions.

Reviewer #3

(Remarks to the Author)

I co-reviewed this manuscript with one of the reviewers who provided the listed reports. This is part of the Nature

Communications initiative to facilitate training in peer review and to provide appropriate recognition for Early Career Researchers who co-review manuscripts.

Version 2:

Reviewer comments:

Reviewer #1

(Remarks to the Author)

Please find the attached file named review_nat_comm_NiPS3__2nd.pdf since our report is therein.

Reviewer #2

(Remarks to the Author)

In the revised manuscript, the authors have fabricated and characterized a monolayer NiPS₃ FET, observing distinct transport behavior compared to FETs based on NiPS₃ nanoflakes thicker than two layers. They propose two-step magnetic phase transitions in 1L NiPS₃: a BKT transition near 120 K and a second transition below 60 K leading to long-range magnetic order. While the revised manuscript shows improvement over the previous version, and the experimental findings in 1L NiPS₃ FET are intriguing, several key questions must be addressed before further consideration for publication:

1. The magnetoresistance measurements in Fig. 3C are conducted along three characteristic directions (aligned with hexagonal crystalline axes, 120° apart) to observe spin-flop behavior. To clarify whether the magnetic symmetry of 1L NiPS₃ is isotropic or hexagonally symmetric, the authors should also examine non-special angles.
2. The data in Fig. 3C were collected at 40 K, where the signal-to-noise ratio is poor. Why were these measurements not performed at lower temperatures (e.g., 5 K, as in Fig. 4a), where noise might be reduced and the spin-flop signal is stronger?
3. A local peak in the dG/dT vs. T curve marks the phase transition near 120 K. Since a second transition occurs around 60 K in 1L NiPS₃, is there any corresponding feature in the dG/dT vs. T curve?
4. The spin-flop field in 1L NiPS₃ (5 T) is significantly lower than in multilayer samples (10 T). Is this due to uniaxial anisotropy in thicker flakes? Could the authors provide theoretical calculations to explain this discrepancy?
5. In Fig. S4b, the spin-flop field remains nearly constant below 50 K before increasing. What causes this apparent threshold behavior?

Reviewer #3

(Remarks to the Author)

Version 3:

Reviewer comments:

Reviewer #2

(Remarks to the Author)

The authors have adequately addressed my questions and comments, and I have no further concerns about this article.

Reviewer #4

(Remarks to the Author)

In this manuscript, the authors have investigated electrical transport behavior of NiPS₃ FET devices with thickness down to monolayer. Both the temperature dependence of conductance and MR results show a significant difference between the monolayer devices and the multi-layer(bulk) devices. Since previous researches applying optical approach can only detect long-range order in NiPS₃ down to two layers, the results of monolayer device are quite impressive. They discover that monolayer NiPS₃ undergoes two magnetic transitions, with a low-temperature phase governed by in-plane hexagonal magnetic anisotropy in contrast with the easy-axis zigzag antiferromagnetic order of the bulk NiPS₃. Generally speaking, this experimental work is well performed and the analysis of their data is rigorous, matching the level and interest of Nature Communications. Before I recommend publish of this manuscript, several of my minor points need to be addressed by the authors:

- 1 The authors have studied the layer-dependence properties of NiPS₃ with electrical methods. Recent studies have investigated the dimensional-crossover behavior in NiPS₃ with optical approach [Z. Sun et al, Nature Physics 20, 764–1771 (2024)] and non-local magnon spin transport method [B. Luo et al, SCIENCE CHINA Physics, Mechanics & Astronomy, 68, 127511 (2025)]. I believe these two articles are highly relevant to the current manuscript. I suggest the author add the discussion about the relationship between the manuscript and these two papers mentioned above.
- 2 The actual order of NiPS₃ is controversial. The bulk NiPS₃ exhibits zigzag antiferromagnetic order with easy axis along the a-axis while below some critical thickness, the NiPS₃ exhibits a three-stage nematic vestigial order [Z. Sun et al, Nature Physics 20, 764–1771 (2024), B. Luo et al, SCIENCE CHINA Physics, Mechanics & Astronomy, 68, 127511 (2025)], which

seems that can also fit with the multi-direction MR results in Fig.3 for the monolayer device. Is the low-temperature phase for monolayer governed by in-plane hexagonal magnetic anisotropy the same as three-stage nematic vestigial order? If yes, the author can use a uniform express as the references, if not, a discussion to distinguish the two phase is needed.

3 The statement in line 123 “Application of external magnetic field along this axis results in a sudden 90° rotation of a Néel vector at a critical field value B_{sf} – a spin-flop metamagnetic transition.” is true for many commonly-seen easy-axis antiferromagnet but not accurate for NiPS₃. There are experimental signatures that for the easy-axis zigzag antiferromagnetic order in bulk NiPS₃, a spin-flop transition tends to pin the Néel vector along the a-axis, irrelevant of the direction of the magnetic field. [X. Wang et al, Nature Communications 15, 8011 (2024), B. Luo et al, SCIENCE CHINA Physics, Mechanics & Astronomy, 68, 127511 (2025)]. The statement can be changed into “Application of external magnetic field results in a reorientation of Néel vector at a critical field value B_{sf} – a spin-flop metamagnetic transition.” which is more universal and rigorous.

4 The monolayer NiPS₃ structure exhibits D_{3d} symmetry, while bulk NiPS₃ adopts a monoclinic stacking configuration with reduced C_{2h} symmetry. The layer-dependent magnetic anisotropy is relevant with the crystal symmetry. More discussion about this is also welcome.

Report on “Nature of 2D XY antiferromagnetism in a van der Waals monolayer”

The authors have investigated the magnetic phase transitions in six-layer (6L), bilayer (2L), and monolayer (1L) NiPS₃. In the case of 6L NiPS₃, they performed polarization-resolved photoluminescence (PL) experiments and transport measurements with and without a magnetic field. They have associated the strong angle-dependent PL spectra with the direction of the Néel vector by fitting their magneto-transport experimental results with a model containing the direction of the Néel vector, i.e., α_{0T} and β_{0T} . In the case of 2L and 1L samples, since PL is not an option, they resorted to transport experiments only and tried a few different directions of the Néel vector in their model to fit the experimental results.

Importantly, they observed the same kink in dG/dT vs T graph from the longitudinal transport measurements without an external magnetic field in all three cases at around similar Néel temperatures: 6L (at 149 K), 2L (at 142.1 K), and 1L (at 139.3 K) samples. Then, from the magneto-transport experiments, they claimed the signature of a spin-flop transition in the 6L and 2L samples but not in the 1L case.

From these results, they conclude that the magnetic phase transition persists all the way down to 1L NiPS₃ but the magnetic phase transition in the 1L case is not the usual one but the BKT transition.

The problem at hand, magnetism in two dimensions, is of fundamental importance. More specifically, whether there is magnetism or not in 1L NiPS₃ is an important controversy nowadays. This paper provides important information for solving this problem to the broad condensed matter physics community. Therefore, we find that the paper meets the high criterion of importance for Nature Communications.

However, we cannot recommend the publication of this paper in its current form in Nature Communications because the experimental data do not support the current conclusion. Rather, as we explain in the following, the experimental data point to a different conclusion, which will interest the broad readership of Nature Communications.

1. Most importantly, the variation in the data presented in Fig. 4d (and also in Fig. S13b) amount to as large as 0.4 %, i.e., higher than 0.2 %, the variation in the magnetoresistance (MR) signal claimed by the authors if the signal was coming from three magnetic domains having their Néel vectors rotated by 120 degrees. This variation could come from some random noises or the actual MR variation. In either case, we cannot conclude that there is no long-range order in 1L samples.

2. Moreover, there are certain overall trends which seem different from random noise in the data shown in Fig. 4d.

(1) Take a look at the curve for B perpendicular to Edge #4 (the fourth curve from the top). Here, we definitely can see a maximum at $B = 0$ and a monotonic field-dependent decrease in the MR for both signs of the magnetic field. Importantly, the MR becomes lower than -0.3 % at $B \sim 10$ T. The magnitude of this value is much larger than that of -0.1 %, the MR value for 2L NiPS₃ if a magnetic field of magnitude 10 T is parallel to Edge #2 as shown in Fig. 4c (the middle curve).

(2) Many features in the data shown in Fig. 4d do not look like random noises. For example, the decrease in MR around $B = -9$ T as the magnetic field becomes further negative.

(3) It is very important that the readers can compare 2L and 1L results fairly. Right now, the results for 1L NiPS₃ shown in Fig. 4d are drawn in 4-times smaller scale than the results for 2L NiPS₃ shown in Fig. 4c, and it is very easy for the readers to mistakenly consider that the signal variation in 1L NiPS₃ is smaller than that in 2L NiPS₃. In fact, the variation in the MR from both samples are of similar magnitude. I believe that this is an honest mistake of the authors. However, this kind of mistake is critical since this comparison is so important. Please enlarge Fig. 4d to the same scale as Fig. 4c. The authors can use two columns for this panel. Also, for a fair comparison, the authors should present data for the magnetic field varying from -14 T to $+14$ T in Fig. 4d as done in Fig. 4c. Please do everything in Fig. 4d in precisely the same way as in Fig. 4c. Also, Figs. 4c and 4d currently do not specify V_{BG} and V_{SD} in each panel. Please specify them in the figure so readers can not miss any essential information.

(4) In the 1L results shown in Fig. S14c, the variation in MR for this 1L sample, $\sim 2\%$, is even much more significant than both the model calculations on 1L NiPS₃ assuming three types of domains and the variation in MR of the 2L samples. Moreover, as in the other 1L sample case shown in Fig. 4d, here we can see certain features in the MR that cannot be assigned to random noises. Therefore, we cannot draw the conclusion that there is no long-range magnetic order in 1L NiPS₃ based on the data.

FIG. 1. Simulated MR of 1L NiPS₃ composed of three domains according to Eq. (S7) of the supplementary information. Note that these results agree well with the results shown in Fig. S15d. We have specified the α_{0T} and β_{0T} values for one domain. The other domains are related to this domain by three-fold rotations.

3. The anisotropic magnetoresistance (AMR) value decreases with the number of layers (6L: -0.45% , 2L: -0.37% or -0.39%), which is reasonable. Therefore, the authors should have used an even smaller AMR value in the simulation for 1L NiPS₃, i.e., in the results shown in Fig. S15d. This change will further decrease the simulated MR of 1L NiPS₃. Please explain this fact in the manuscript.

4. Unlike the numerical simulations performed by the authors, in actual experimental situations, there are inhomogeneities in the angles α_{0T} and β_{0T} even if the Néel vector is well-defined in the entire sample globally or small domains locally. These angles are determined by the directions of the Néel vector, the magnetic field, and the current direction, which is determined by the fabricated device electrodes. As we show below, the calculated variation in the simulated MR becomes even smaller and smoother if more realistic parameters are used and the inhomogeneities are reflected.

(1) The β_{0T} value used to simulate the data shown in Fig. 2c for 6L samples is too small, which is why the simulated results are sharper in their variation near the peak ③. The authors should use a larger value of β_{0T} , e.g., $\beta_{0T} = 5^\circ \sim 10^\circ$ that best describes the experimental data.

(2) In the simulation of 1L NiPS₃ composed of three domains, they assumed certain α_{0T} and β_{0T} . However, as we show below, the results are sensitive to these parameters and the variation in the simulated MR can become even smaller and smoother if slightly different parameters are used. First, we show the validity of our simulation based on Eq. (S7) of the supplementary information. Figure 1 of this report reproduces the results shown in Fig. S15d, thus validating our calculations.

(3) Figure 2 of this report shows that the MR in 1L NiPS₃ is very sensitive to the β_{0T} value. The authors should show these results to the readers so they know that the variations in the MR with respect to the magnetic field become smoother and smaller if a larger β_{0T} is used. Note that the used value of $\beta_{0T} = 2^\circ$ is unrealistically small, as pointed out in the case of 6L NiPS₃. Therefore, the authors should replace the data shown in Fig. S15d with the results obtained with $\beta_{0T} = 5^\circ \sim 10^\circ$. At the same time, use a smaller value of AMR in the new simulation to reflect that the AMR is a decreasing function of the number of layers.

(4) Figure 3 of this report shows that the MR in 1L NiPS₃ is also very sensitive to the α_{0T} value. As the authors mention, there are uncertainties about whether the crystallographic zigzag direction is parallel or perpendicular to one of the sample edges. Also, because the fabricated electrodes determine the current direction, α_{0T} cannot be a homogeneous quantity. It should vary within a single domain and the entire sample. For these two reasons, the authors should present these results to the readers and let them know that the variation in the MR can become smaller, smoother, and qualitatively different from the results shown in current Fig. S15d. At the same time, use a smaller value of AMR in the new simulation to reflect that the AMR is a decreasing function of the number of layers.

5. Last but not least, an important fact is that there is a magnetic phase transition in 1L NiPS₃ whose signature in longitudinal transport experiments is qualitatively the same as that of multilayer systems. Remarkably, this kink feature amounts to $\sim 15\%$ of dG/dT (Fig. 4b). This significant result should be emphasized in the main text. Currently, it is almost overlooked. How come should we conclude that this qualitatively the same kink feature in the dG/dT vs T of 1L NiPS₃ comes from the BKT transition whereas the kink for 6L or 2L NiPS₃ comes from the

FIG. 2. Simulated MR of 1L NiPS₃ composed of three domains according to Eq. (S7) of the supplementary information with different values of β_{0T} for one domain. The other domains are related to this domain by three-fold rotations.

ordinary magnetic phase transition accompanying long-range order? We should not. The conclusion should be that they observed qualitatively the same behavior in dG/dT measurement from 6L all the way down to 1L samples, i.e., the presence of magnetic phase transitions in 1L NiPS₃. The paper should not arrive at a hasty conclusion that there is no long-range order in 1L NiPS₃.

For the above reasons, we cannot recommend publishing this paper in Nature Communications in its current form. The authors are advised to make the modifications described so far and change the title and the abstract properly so that the paper does not mislead the readers.

The following are other important issues to be addressed by the authors to improve the manuscript.

1. β_{0T} is the angle between the direction of the applied magnetic field and the direction of the Néel vector. The authors should specify the direction of the applied magnetic field for each MR measurement in each figure panel. The readers know that there are uncertainties in the Néel vector direction. Still, at least the other part, the magnetic field direction, is an independent variable that can be set in each experiment. This direction should be specified as an angle value (and as an arrow when appropriate in each panel) in all the figures in the main manuscript and the supplementary information. Also, analyze the error in the direction of the in-plane magnetic field of the experimental setup. How large is the uncertainty in the magnetic field's direction (in-plane angle)? This uncertainty will determine part of the error in β_{0T} . (The other part comes from the uncertainty and inhomogeneity of the Néel vector direction.) Please explain the experimental setup to support the authors' estimation. The readers should know very clearly the sources of the error for β_{0T} and how large each of them is.

2. Please show experimental results for an extensive scan of the β_{0T} value at grid points, including the results for large β_{0T} values; for example, in steps of 5 degrees, for 6L, 2L, and 1L samples. Note that the magnetic field direction is not fixed by the device fabrication, such as electrodes, and can be changed in the experiment. Then, present

FIG. 3. Simulated MR of 1L NiPS₃ composed of three domains according to Eq. (S7) of the supplementary information with different values of α_{0T} for one domain. The other domains are related to this domain by three-fold rotations.

numerical simulations and the best fit for the experimental data using a single AMR value for each sample. This comprehensive comparison between the experimental results and the simulation is required to assess the validity of the AMR theory used in the simulation. Currently, the readers might think that the paper intentionally cherry-picks a tiny portion of the available phase space of the parameters for the MR measurement.

3. A crucial thing in analyzing the MR data using the AMR model is to fix the AMR overall constant and α_{0T} for a given sample and to fix β_{0T} for one magnetic field direction for this sample and calculate β_{0T} at other magnetic field directions using these parameters because the direction of the magnetic field is tunable and is known; that is, the authors should not fit β_{0T} at each given magnetic field direction. Suppose that they fit $\beta_{0T} = (\text{angle of the magnetic field}) - (\text{angle of } \mathbf{L})$. Then, one obtained (angle of \mathbf{L}) from fitting. Using this parameter, one can calculate $\beta_{0T} = (\text{angle of the magnetic field}) - (\text{angle of } \mathbf{L})$ at magnetic fields pointing along different directions.

4. In all samples (6L, 2L, 1L) and at all field strengths, the MR is always negative. However, their interpretation does not prevent positive MR. In fact, the two are equally observable according to Eqs. (1) and (2) of the main manuscript. To make people believe their AMR interpretation, showing results with positive MR is essential. The important parameters that determine the sign of the MR are the angles between the magnetic field, the Néel vector, and the current direction. The angle between the Néel vector and the in-plane magnetic field direction can be changed easily, and the angle between the Néel vector and the current direction is determined by the device setup. They should provide the results for 6L, 2L, and 1L samples with positive MR.

5. Please provide similar transport (dG/dT vs T) and the MR measurements for a bulk (or a much thicker) sample; these additional data would be very helpful in understanding the trend of how dG/dT and the MR change from bulk all the way down to 1L NiPS₃.

6. The authors cite Ref. 22 as follows: “Consistently with these considerations, recent Monte Carlo simulations of magnetism in 1L NiPS₃ confirm the occurrence of a BKT phase transition with $T_{\text{BKT}} = 141 \text{ K}$.” However, the model considered in Ref. 22 assumes zero in-plane anisotropy, which is not a valid assumption according to the advanced first-principles calculations reported in Ref. 21. Such a model without in-plane anisotropy, in disagreement with first-principles calculations, will of course lead to a BKT transition. Note that first-principles studies like Ref. 21 have quantitatively reproduced the magnetic anisotropy of van der Waals magnetic materials: for example, Xu et al., “Interplay between Kitaev interaction and single ion anisotropy in ferromagnetic CrI₃ and CrGeTe₃ monolayers,” npj Comput. Mater. **4**, 57 (2018), have reported results that are in quantitative agreement with the tiny magnetic anisotropy of 2D CrGeTe₃ [Nature **546**, 265 (2017)] and large easy-axis anisotropy (along out-of-plane direction) of 2D CrI₃ [Nature **546**, 270 (2017)] measured from experiments. Therefore, the authors should explain these details in citing Refs. 21 and 22 and cite Xu et al. 2018 npj Comput. Mater. paper together with the two back-to-back 2017 Nature papers in this context.

7. The authors mention that the absence of the in-plane easy-axis anisotropy can be explained by the structural symmetry of 1L NiPS₃. This is not true because even if the underlying structure retains the trigonal symmetry, the zigzag magnetic pattern, which is known to be the lowest-energy configuration of the *isotropic* J_1 - J_2 - J_3 model of NiPS₃, breaks the (trigonal) symmetry of the charge and spin densities and in turn induces an easy-axis anisotropy through spin-orbit coupling. Therefore, the authors should explain in the main manuscript that the widely-accepted, lowest-energy magnetic configuration for NiPS₃ does NOT allow the lack of in-plane easy-axis anisotropy in 1L NiPS₃.

8. Because the system is not a 1D material which can be thought of as serially connected domains of different resistances but a 2D material which is effectively a 2D network of resistances, a simple average of the MR in three different domains, Eq. (S10), which works for a 1D system, will make an additional error from the total MR of the mosaic sample composed of patches of three different domains. Please mention the limitation of Eq. (S10) in the manuscript.

9. In all figures showing the MR results in both the main manuscript and in the supplementary information, show the results for the magnetic field from -14 T to $+14 \text{ T}$, instead of from -12 T to $+12 \text{ T}$ as done in some figures. Also, for example, in Figs. S13 and S14, provide $B < 0$ regime as well up to -14 T , as done for other cases.

10. In all figures showing simulated MR results in both the main manuscript and in the supplementary information, specify the α_{0T} , β_{0T} , and the AMR values used in the simulations.

11. Why does the noise in MR increase if the temperature decreases? This trend is observed in all 6L, 2L, and 1L samples. Please add an explanation of this phenomenon to the main manuscript.

12. To Fig. 4b, please also add the results for 6L. Moreover, raw experimental data for G and dG/dT should be provided. If there is a space problem, supplementary information is a good option. Currently, only $G/G_{200\text{K}}$ is presented.

13. Please specify V_{BG} and V_{SD} in each panel of the figure or the caption for all the figures in the main manuscript and the supplementary information. Readers should easily understand the experimental setup for the results shown in each figure without reading the relevant part of the manuscript.

14. In line 207, “which was taken as evidence for the absence of magnetic order” should read “which was suggested as the origin of the absence of a particular peak splitting in the Raman spectrum even if the ground state is magnetic” because Ref. 21 did not claim the absence of magnetic phase transition in 1L NiPS₃; indeed, the opposite was supported in Ref. 21, i.e., the presence of magnetization in 1L NiPS₃.

15. In Fig. S10, the best AMR value for 2L NiPS₃ is -0.37% , but in line 186 of the main text, it is said to be -0.39% . Is this a proper comparison? If so, which value is correct?
16. After Eq. (3) of the main manuscript, please explain the relation between K and D^x .
17. The following sentence between Eq. (S3) and Eq. (S4) is not correct. Please revise it: “ \mathbf{L} and \mathbf{M} satisfy $\mathbf{L} \cdot \mathbf{J} = 0$.”
18. In line 183, ‘edges #1 and #2’ should read ‘edges #2 and #3,’ and in lines 185 and 189, ‘edge #3’ should read ‘edge #1.’ Please check these things.
19. In the reference list of the main text, Ref. 8 is the same as Ref. 21. Please delete one of them.

Report on “Nature of 2D XY antiferromagnetism in a van der Waals monolayer”

Based on the criticisms and comments of the referees, the authors have significantly revised the manuscript. In particular, they said that what they had thought to be a monolayer (1L) sample was actually a bilayer (2L) and that they had found true 1L samples. Moreover, they abandoned their theoretical model on the spin-flop transition because the model does not work very well for 1L and 2L samples.

The authors have also entirely changed their central claim. Now they say that there is indeed a magnetic phase transition in their 1L sample, albeit at a lower temperature ($T^* \approx 60$ K). They bring an old paper [Ref. 42, which is J. V. José *et al.*, Phys. Rev. B **16**, 1217 (1977)] to explain that there are two phase transitions: a BKT transition at $T_c = 124$ K and a normal magnetic phase transition at $T^* = 60$ K, below which a long-range order is formed.

We appreciate the authors for performing additional experiments and trying to clarify the issues. However, for the following reasons, we find that the new, completely different manuscript has its own serious problems. We still believe that the subject is of importance to the general readership of Nature Communications; however, the conclusion is not supported by the experimental data, as we explain in detail. Therefore, we cannot recommend the publication of this paper in Nature Communications in its current form. However, we think that the authors may be given one last chance to revise the paper so that they reach a conclusion consistent with their experimental data and provide their high-quality experimental data and possible explanations.

1. First of all, the authors provide no experimental or theoretical evidence to support their claim that what they observed for the magnetoresistance (MR) of 1L NiPS₃ is what was proposed by the José et al.’s paper other than the fact that, according to the authors’ claim, only 1L NiPS₃ shows some two-step-like temperature dependence in the measured MR, although we also doubt this claim about 1L NiPS₃ as we detail later. They do not compare the detailed contents of the paper by José et al. or any other theoretical paper with their experimental data (conductance at $B = 0$ and MR). Importantly, the paper by José et al. is on the XY model with hexagonal symmetry-breaking term at $B = 0$; however, the measured conductance at $B = 0$ as a function of temperature does not show the two-temperature feature that could possibly be deduced from the model. (We can only say this weakly because the paper by José et al. did not make any predictions on the conductance even for the $B = 0$ case.) Still, the authors apply the paper by José et al., which is on the $B = 0$ case only, to their MR data, i.e., experimental data performed under a non-zero magnetic field. Obviously, this inconsistent, cherry-picking comparison is groundless. Moreover, their experimental data for $B = 0$ can be explained quantitatively by first-principles calculations assuming $B = 0$, as we explain in our next point. However, we do not blame the authors for this complete absence of evidence because this subject actually belongs to uncharted territory, as we also explain in our next point.

2. José et al.’s paper assumes a simple hexagonal symmetry-breaking term of the form $\cos(6\theta)$ in addition to the perfect XY Hamiltonian. However, this model does NOT reflect the nature of the effective Hamiltonian for NiPS₃. Indeed, according to first-principles calculations [Ref. 19, which is Nano Lett. **21**, 10114 (2021)], the ground state of NiPS₃ is six-fold degenerate, but for a different reason: three degenerate zigzag chain directions separated by an angle of $2\pi/3$ for the antiferromagnetic spin configuration and two options for the collinear spin alignment along the given zigzag chain direction (the two ground-state options are connected by inverting all the spins). However, this 6-fold degeneracy of NiPS₃ is qualitatively different from the simple, naive $\cos(6\theta)$ term of Ref. 42. According to the state-of-the-art first-principles calculation in Ref. 19, this 1L NiPS₃ (hosting 6-fold-degenerate ground states) undergoes a magnetic phase transition, and the calculated T_c drop of 3 K in going from 3L to 2L and that of 15 K in going from 2L to 1L agree perfectly with the measured values of 3 K and 16 K, respectively. Therefore, the results on the conductance at $B = 0$ could indeed be strong quantitative evidence of qualitatively the same magnetic phase transition in NiPS₃ from bulk all the way down to 1L samples in agreement with state-of-the-art first-principles calculations assuming zero magnetic field. This is a beautiful experimental verification of the prediction from state-of-the-art first-principles calculations. To our knowledge, we have not seen a plot like Fig. 1d in any previous studies. This excellent agreement between theory and experiment should be mentioned and emphasized in the manuscript.

3. If the experimental results provide evidence for the existence of the magnetic, ordered phase in NiPS₃ at $B = 0$ from bulk all the way down to monolayer samples, then the natural question arising would be how to explain the experimental results with a non-zero in-plane magnetic field from bulk all the way down to monolayer NiPS₃. So far, nobody knows about the temperature-dependent MR and spin-flop transitions in 1L NiPS₃, which hosts such complicated 6-fold degenerate ground states (qualitatively different from the simple model in Ref. 42, unsuitable for

1L NiPS₃). However, this subject is far beyond the scope of this article reporting on an experimental study. This paper beautifully provides experimental evidence for the magnetic phase transition in 1L NiPS₃ at $B = 0$, qualitatively the same as in thicker samples; If published in Nature Communications, this paper will lead to theoretical studies on this important subject, the MR and spin-flop transitions of 1L NiPS₃, which hosts the exotic 6-fold degenerate ground states based on the 3-state nematicity, under an in-plane magnetic field.

4. As we explained before, the significant drop in T_c in going from 2L to 1L NiPS₃ ($T_c^{2L} - T_c^{1L} = 16$ K) in comparison with the drop in T_c in going from 3L to 2L ($T_c^{3L} - T_c^{2L} = 3$ K) is a quantitative difference, not a qualitative one. This quantitative difference is due to the strong interlayer spin exchange interactions and is unique for NiPS₃. Other compounds like MnPS₃ or FePS₃ do not show such a significant change in T_c in going from 2L to 1L because their interlayer exchange interactions are much weaker than those of NiPS₃, as revealed and explained by first-principles calculations in Ref. 19. The authors should discuss the effect of unusually strong interlayer exchange interactions in NiPS₃ in the main manuscript.

In this way, we may be able to understand many of the experimental results that were assigned to two-step magnetic phase transitions. (i) For example, the 1L data obtained at $T = 40$ K presented in Fig. 3d shows a sharp decrease in MR at $B \sim 5$ T. In contrast, the MR of 1L NiPS₃ at $T = 5$ K shown in Fig. 4a drops sharply around $B = 7$ T and resembles the 2L data at $T = 40$ K in Fig. 3d. A closer examination of the 2D scan of the MR of 1L NiPS₃ shown in Fig. 4c reveals that the magnetic field strength at the onset of the sharp decrease in MR reaches a minimum around ~ 33 K, which is close to the temperature ~ 40 K at which the 1L data in Fig. 3d was obtained. In this context, we can understand some of the differences between 1L and 2L MR data as being quantitative, rather than qualitative. Please revise the manuscript in light of this observation. (ii) Also, it is clear that in going from 6L to 2L, B_{sf} (spin-flop field) at $T = 90$ K decreases from 11.5 T (Fig. 2e or Fig. S4b) to 10.7 T (Fig. S7c).

How these trends evolve in samples of different thicknesses could provide some critical clues and additional experimental data that are valuable for future studies. Currently, the authors provide a detailed analysis of 6L and 1L samples only. They do not provide any 2D scan for 3L, and, in the case of 2L sample, they provide the 2D scan only in a very narrow temperature range (Fig. S7b). The readers should be able to view the complete experimental data and know the trend on how the 2D scan varies from 13L to 6L, to 3L, to 2L, and to 1L NiPS₃. We request the authors to provide the following data in the Supplementary Information.

(1) Provide complete 2D scans of magnetoresistance (MR) vs temperature and magnetic field for T from 0 K to 180 K and B from -12 T to $+12$ T (see panel a of Fig. 1 of this report) for all the samples: 13L, 6L, 3L, 2L, 2L, 2L (, 2L, 2L) (they have three or five different 2L samples), 1L, 1L, 1L (the authors also have three different 1L samples as shown in Fig. S2). Also, we suggest that the authors provide more 6L, 5L, 4L, 3L, and, most importantly, more 1L samples and provide the same 2D scans. These additional samples will be very important for the readers to distinguish which features are universal and which features depend on the quality of the sample and the device configurations, such as V_{SD} and V_{BG} .

(2) For the 2D scans, use the same color bar in all cases (samples of different thicknesses) so that the readers can make an unbiased, fair comparison by themselves.

(3) For each 2D scan of MR, add a dR/dB 2D scan as well, as done in Fig. S4a. See panel b of Fig. 1 of this report. Use the same color map in all dR/dB 2D scans, also for samples with different thicknesses.

(4) For each of the 2D scans, add a graph showing B_{sf} vs. T (see panel c of Fig. 1 of this report). For each T , there should be two B_{sf} values, one positive and one negative. The range for B_{sf} should also be from -12 T to 12 T, and that for T should be from 0 K to 180 K. We know that there will be no data points at high temperatures; however, for the readers to be able to fairly compare samples of different thicknesses, it is very important that all the graphs (corresponding to different thickness samples) show exactly the same range of B_{sf} and T .

(5) Very importantly, for each data point in the graph described in (4), provide the MR vs. B graph (see panel d of Fig. 1 of this report) so that the readers can know how they extracted B_{sf} from the raw MR data. (Also, explain how B_{sf} was chosen in the revised manuscript.) If there are too many curves, they can split the panel into a few panels but should not leave out any curves.

(6) Regarding (2) and (3), please provide additional (separate) figure(s) with different color bar(s) if the authors judge that using a single universal color bar for all samples may miss some essential features. However, in this case, the authors should also show all the sample results in the additional figure(s) using the same color bar in each additional figure. So, the authors may make 2 to 3 figures in total: Each figure shows all sample 2D scans with the same color

bar. However, the plots for B_{sf} vs. T and the plots for MR vs. B (panels c and d of Fig. 1 of this report, respectively) do not need to be repeated in the additional figure(s).

(7) A closer look at Fig. 4b shows that the temperature at which dG/dT becomes negative closely matches T^* ($= 60$ K). Interestingly, if we estimate the temperatures at which dG/dT turns negative in Fig. 1c for various thicknesses, we find that the temperatures are around 80 K and 100 K for 2L and 6L samples, respectively. These temperatures are similar to those below which the sharp features in the MR at $B \sim 8\text{--}10$ T appear in each case (see Fig. S7a and Fig. 2c, respectively). Is this correlation that appears across multiple datasets in this paper a coincidence? Provide additional data to judge whether the three 1L devices presented in Fig. S2 all show the switching behavior at the same temperature.

(8) In addition, thoroughly investigate how the 2D scans depend on V_{SD} and V_{BG} for samples of different thicknesses, including the 1L, 1L, 1L, 2L, 2L, 2L, (2L, 2L,) and 6L samples.

5. B_{sf} is a rapidly increasing function of temperature. Therefore, we do not know if there will be a spin-flop transition in 1L NiPS₃ at higher temperatures if higher magnetic fields are applied. The authors should explain this temperature dependence of B_{sf} and the possibility of observing spin-flop transitions in 1L NiPS₃ at higher temperatures under higher magnetic fields in the main manuscript.

6. The sentence in the abstract “While bilayer and multilayer NiPS₃ exhibit a single magnetic phase transition into a zig-zag antiferromagnetic state driven by uniaxial anisotropy, monolayer NiPS₃ undergoes two magnetic transitions, with a low-temperature phase governed by in-plane hexagonal magnetic anisotropy” is inconsistent with the data presented in this paper. The temperature below which the spin-flop transition starts to occur and T_c are also different in the cases of 2L, 3L, 6L, and 13L samples. For example, for the 6L sample, a spin-flop transition is observed for B within $-12 \sim 12$ T only up to 100 K (see Fig. 2d and Fig. S4a), which is significantly lower than $T_c \sim 149$ K. Please explain this difference between the relevant temperatures for magnetic phase transitions and spin-flop transitions in samples thicker than 1L NiPS₃ in the abstract and also in other parts of the paper that have similar claims.

7. The sentences “We recall that the magnetic structure of bulk NiPS₃ is governed by a strong easy-plane anisotropy ($D_z \approx 0.2$ meV per Ni), which confines the spins to predominantly lie in the ab-plane, and a weak uniaxial anisotropy ($D_x \approx -0.01$ meV per Ni). The latter is believed to originate from interlayer interactions due to the monoclinic layer stacking, and orients the spins (and hence the Néel vector, L) along the crystallographic a-axis (Figure 1a)” is not correct. The zigzag anti-ferromagnetic spin configuration can also significantly affect the direction of the spin polarization. The magnetic anisotropy arising from the intralayer interactions can be an order of magnitude larger than that from the interlayer interactions as explained in Ref. 19. The authors should revise the explanation.

8. The 2L data in Fig. 1c and Fig. S2b of the current version and Fig. 4b of the previous version (both the 2L and the 1L data, since the authors now claim that the 1L sample was actually also a 2L one) are very different from each other. Please explain why any currently presented 2L data are different from the previous data (for 2L or 1L - which is supposedly 2L - samples). Please show (1) all 3 curves in Fig. S2b and the 2L data (and also the 1L data because it was actually also a 2L sample) of the previous version in Fig. 1c of the revised manuscript (i.e., five different curves together).

9. For the 6L sample, the sharp spin-flop transition signatures do NOT exist near 150 K; rather, the signature is visible only below 110 K. Should we interpret the results so that there are also two different phase transitions in 6L NiPS₃? We do not think so. As we explained before, the authors should provide the 2D scan as in Fig. 2d and Fig. 4c for all the other samples, 13L, 3L, 2L, etc. Also, provide 2D scans for different samples of the same thickness, i.e., three (or five, including the ones from the previous version of the manuscript) such 2D scans for the 2L samples and three 2D scans for the 1L samples, since the authors already have them. If the authors have more 13L, 6L, and 3L samples, they are advised to show the 2D scans for them as well. The readers should be able to compare these thorough 2D scans before drawing any conclusions.

10. Similarly to the results in Figs. 3b and 3c, what do the results look like if the magnetic field is perpendicular to the edges of 2L and 1L NiPS₃ samples? Please provide figures like Figs. 3b and 3c for the case where the magnetic field is perpendicular to the edges for all three (or five) 2L samples and all three 1L samples.

FIG. 1. Schematic of the 2D scan of MR to be added to the Supplementary Information.

Response to reviewers' critique for the manuscript “*Nature of 2D XY antiferromagnetism in a van der Waals monolayer*” by Cheol-Yeon Cheon, Volodymyr Multian, Kenji Watanabe, Takashi Taniguchi, Alberto F. Morpurgo, and Dmitry Lebedev.

Reply to all reviewers:

We would like to thank the reviewers for providing critical feedback. After carefully reviewing the comments we decided to conduct more measurements of 1L NiPS₃ and upon averaging observed a clear MR signal, characteristic of 2L (i.e. spin-flop above 10 T). We therefore re-examined the optical contrast of exfoliated few-layer NiPS₃ (shown now in **Figure S9**) and found that the optical contrasts of the previously identified 1L and 2L fall within the error range of the second-lowest contrast value ($8 \pm 1\%$). This observation along with the very close transition temperatures from dG/dT measurements lead to the conclusion that the device we initially identified as 1L is in fact 2L.

We therefore prepared and characterized actual 1L NiPS₃ FETs and showed that their behavior is drastically different from thicker samples. In particular, we found that 1L NiPS₃ undergoes two distinct magnetic phase transitions: one at $T_c = 124$ K revealed from conductance derivative dG/dT (which is significantly lower than approx. 140 K for 2L) and another at $T^* = 60$ K, seen by the onset of MR and spin-flop as well as two-state fluctuations of electrical current. Below the T^* , we observe spin-flop transitions when magnetic field is aligned along all three zig-zag edges 120° apart from each other, contrary to thicker samples, where spin-flop transition is only observed when field is applied along the easy axis. Moreover, the spin-flop transition in 1L NiPS₃ occurs at much lower critical field of approx. 5 T, as compared to 10 T for 2L and thicker.

These experimental observations allow us to conclude that low-T phase of 1L NiPS₃ ($T < T^* = 60$ K) is characterized by hexagonal magnetic anisotropy, unlike uniaxial anisotropy of 2L and thicker NiPS₃ induced by monoclinic layer stacking. We identify the intermediate magnetic phase ($60 < T < 124$ K) as 2D XY phase with absence of long-range magnetic order. Such behavior matches the renormalization group calculations of 2D XY model with symmetry-breaking anisotropy terms. According to these calculations, a system with weak hexagonal in-plane anisotropy is expected to undergo two phase transitions: Berezinskii–Kosterlitz–Thouless (BKT) transition to a state without long-range order at a higher temperature $T = T_{\text{BKT}}$ and spontaneous symmetry breaking, establishing long-range order with the Néel vector along one of six symmetry-equivalent degenerate directions.

Based on these new results, we have deeply revised the text and extended our discussion. We shifted the emphasis of the manuscript from anisotropic magnetoresistance measurements and modelling (which become topic of a separate report) to constructing the phase diagram of 1L NiPS₃ and thus experimental validation of theoretical predictions of 2D XY model with symmetry-breaking fields. We thank the reviewers for critical analysis of our work and below we provide a point-by-point response to the comments raised during the first stage of review.

Reply to Reviewer #1 :

Comment 1:

The authors have investigated the magnetic phase transitions in six-layer (6L), bilayer (2L), and monolayer (1L) NiPS₃. In the case of 6L NiPS₃, they performed polarization-resolved photoluminescence (PL) experiments and transport measurements with and without a magnetic field. They have associated the strong angle-dependent PL spectra with the direction of the Néel vector by fitting their magneto-transport experimental results with a model containing the direction of the Néel vector, i.e., α_{0T} and β_{0T} . In the case of 2L and 1L samples, since PL is not an option, they resorted to transport experiments only and tried a few different directions of the Néel vector in their model to fit the experimental results.

Importantly, they observed the same kink in dG/dT vs T graph from the longitudinal transport measurements without an external magnetic field in all three cases at around similar Néel temperatures: 6L (at 149 K), 2L (at 142.1 K), and 1L (at 139.3 K) samples. Then, from the magneto-transport experiments, they claimed the signature of a spin-flop transition in the 6L and 2L samples but not in the 1L case.

From these results, they conclude that the magnetic phase transition persists all the way down to 1L NiPS₃ but the magnetic phase transition in the 1L case is not the usual one but the BKT transition.

The problem at hand, magnetism in two dimensions, is of fundamental importance. More specifically, whether there is magnetism or not in 1L NiPS₃ is an important controversy nowadays. This paper provides important information for solving this problem to the broad condensed matter physics community. Therefore, we find that the paper meets the high criterion of importance for Nature Communications.

However, we cannot recommend the publication of this paper in its current form in Nature Communications because the experimental data do not support the current conclusion. Rather, as we explain in the following, the experimental data point to a different conclusion, which will interest the broad readership of Nature Communications.

1. Most importantly, the variation in the data presented in Fig. 4d (and also in Fig. S13b) amount to as large as 0.4 %, i.e., higher than 0.2 %, the variation in the magnetoresistance (MR) signal claimed by the authors if the signal was coming from three magnetic domains having their Néel vectors rotated by 120 degrees. This variation could come from some random noises or the actual MR variation. In either case, we cannot conclude that there is no long-range order in 1L samples.

Reply 1: We would like to thank the reviewer for acknowledging the importance of our work. As mentioned in the reply to all the reviewers, we identified new monolayer NiPS₃ flakes, and fabricated new 1L FETs, whose studies became central for the updated version of the manuscript. Magnetoresistance measurements of new 1L NiPS₃ shows clear signal above the noise threshold (e.g., the noise level (standard deviation) of new 1L NiPS₃ is about 0.02% at T=5K, which is an order of magnitude smaller than the MR signal at B=10T; see Fig. 4b for details).

Comment 2:

2. Moreover, there are certain overall trends which seem different from random noise in the data shown in Fig. 4d.

(1) Take a look at the curve for B perpendicular to Edge #4 (the fourth curve from the top). Here, we definitely can see a maximum at $B = 0$ and a monotonic field-dependent decrease in the MR for both signs of the magnetic field. Importantly, the MR becomes lower than -0.3% at $B \sim 10$ T. The magnitude of this value is much larger than that of -0.1% , the MR value for 2L NiPS₃ if a magnetic field of magnitude 10 T is parallel to Edge #2 as shown in Fig. 4c (the middle curve).

(2) Many features in the data shown in Fig. 4d do not look like random noises. For example, the decrease in MR around $B = -9$ T as the magnetic field becomes further negative.

Reply 2: These data are replaced with the new data for 1L NiPS₃ shown on Fig. 3 and Fig. 4.

Comment 3 :

(3) It is very important that the readers can compare 2L and 1L results fairly. Right now, the results for 1L NiPS₃ shown in Fig. 4d are drawn in 4-times smaller scale than the results for 2L NiPS₃ shown in Fig. 4c, and it is very easy for the readers to mistakenly consider that the signal variation in 1L NiPS₃ is smaller than that in 2L NiPS₃. In fact, the variation in the MR from both samples are of similar magnitude. I believe that this is an honest mistake of the authors. However, this kind of mistake is critical since this comparison is so important. Please enlarge Fig. 4d to the same scale as Fig. 4c. The authors can use two columns for this panel. Also, for a fair comparison, the authors should present data for the magnetic field varying from -14 T to $+14$ T in Fig. 4d as done in Fig. 4c. Please do everything in Fig. 4d in precisely the same way as in Fig. 4c. Also, Figs. 4c and 4d currently do not specify V_{BG} and V_{SD} in each panel. Please specify them in the figure so readers can not miss any essential information.

Reply 3: As suggested by the reviewer, we are now plotting the MR results of new 1L with those of other thicknesses using the same scale for MR (%) and magnetic field range (Fig. 3). Additionally, we've also specified the values of V_{BG} and V_{SD} in each caption.

Comment 4 :

(4) In the 1L results shown in Fig. S14c, the variation in MR for this 1L sample, $\sim 2\%$, is even much more significant than both the model calculations on 1L NiPS₃ assuming three types of domains and the variation in MR of the 2L samples. Moreover, as in the other 1L sample case shown in Fig. 4d, here we can see certain features in the MR that cannot be assigned to random noises. Therefore, we cannot draw the conclusion that there is no long-range magnetic order in 1L NiPS₃ based on the data.

Reply 4: For our new data we concluded that the long-range magnetic order in 1L NiPS₃ occurs below 60 K. We observe a clear spin flop transition above the noise threshold.

Comment 5: 3. *The anisotropic magnetoresistance (AMR) value decreases with the number of layers (6L: -0.45% , 2L: -0.37% or -0.39%), which is reasonable. Therefore, the authors should have used an even smaller AMR value in the simulation for 1L NiPS₃, i.e., in the results shown in Fig. S15d. This change will further decrease the simulated MR of 1L NiPS₃. Please explain this fact in the manuscript.*

Reply 5: In the updated version of the manuscript, we decided to focus on magnetic behavior of 1L NiPS₃ and have therefore removed the discussion of the MR mechanism in a few-layer NiPS₃ which will become a topic of a separate report. For this reason, we have removed the AMR simulation of 1L NiPS₃.

Comment 6

4. *Unlike the numerical simulations performed by the authors, in actual experimental situations, there are inhomogeneities in the angles α_{0T} and β_{0T} even if the Néel vector is well-defined in the entire sample globally or small domains locally. These angles are determined by the directions of the Néel vector, the magnetic field, and the current direction, which is determined by the fabricated device electrodes. As we show below, the calculated variation in the simulated MR becomes even smaller and smoother if more realistic parameters are used and the inhomogeneities are reflected. (1) The β_{0T} value used to simulate the data shown in Fig. 2c for 6L samples is too small, which is why the simulated results are sharper in their variation near the peak 3. The authors should use a larger value of β_{0T} , e.g., $\beta_{0T} = 5^\circ \sim 10^\circ$ that best describes the experimental data.*

Reply 6: Although we decided to dedicate a separate manuscript to the AMR modelling of NiPS₃ magneto-transport, here we would like to comment on aligning our magnetic field, as it is still relevant for the current manuscript. To extract the misaligning angle, we followed the reviewer's suggestions and carefully re-examined our fitting procedure of AMR model to MR data for 6L NiPS₃. We show that $\beta_0 = 3.4^\circ$ provides the best fit and use of larger β_0 values fail to reproduce the experimental data accurately (**Fig. R1**). The small value of $\beta_0 = 3.4^\circ$ is consistent with the precision of aligning in-plane magnetic field in our experimental setup (see Reply 10 for more detail). We have included the estimation for the misalignment error in the methods section of the manuscript.

REDACTED

Comment 7

(2) *In the simulation of 1L NiPS₃ composed of three domains, they assumed certain α_{0T} and β_{0T} . However, as we show below, the results are sensitive to these parameters and the variation in the simulated MR can become even smaller and smoother if slightly different parameters are used. First, we show the validity of our simulation based on Eq. (S7) of the supplementary information. Figure 1 of this report reproduces the results shown in Fig. S15d, thus validating our calculations.*

(3) *Figure 2 of this report shows that the MR in 1L NiPS₃ is very sensitive to the β_{0T} value. The authors should show these results to the readers so they know that the variations in the MR with respect to the magnetic field become smoother and smaller if a larger β_{0T} is used. Note that the used value of $\beta_{0T} = 2^\circ$ is unrealistically small, as pointed out in the case of 6L NiPS₃. Therefore, the authors should replace the data shown in Fig. S15d with the results obtained with $\beta_{0T} = 5^\circ \sim 10^\circ$. At the same time, use a smaller value of AMR in the new simulation to reflect that the AMR is a decreasing function of the number of layers.*

(4) *Figure 3 of this report shows that the MR in 1L NiPS₃ is also very sensitive to the α_{0T} value. As the authors mention, there are uncertainties about whether the crystallographic zigzag direction is parallel or perpendicular to one of the sample edges. Also, because the fabricated electrodes determine the current direction, α_{0T} cannot be a homogeneous quantity. It should vary within a single domain and the entire sample. For these two reasons, the authors should present these results to the readers and let them know that the variation in the MR can become smaller, smoother, and qualitatively different from the results shown in current Fig. S15d. At the same time, use a smaller value of AMR in the new simulation to reflect that the AMR is a decreasing function of the number of layers.*

Reply 7: After analyzing the MR data across various thicknesses down to 1L, we found that our current AMR model explains the MR behavior of thicker NiPS₃ (e.g., 6L and 13L), but requires considerable refinement to accurately capture the observed MR behavior of 2L and 1L NiPS₃. We believe that incorporating a revised AMR model into the current manuscript would detract the reader from the central message we now wish to convey, that is the unique phase diagram of 1L NiPS₃ with 2D XY magnetism. We therefore intend to address the detailed mechanism of AMR in NiPS₃ in our future work.

Nevertheless, we deeply appreciate the reviewer's careful examination of our model and insights into the factors that may lead to qualitatively different response in experiments compared to the simulated MR curves. We are seriously taking these considerations in our ongoing AMR study.

Comment 8: *5. Last but not least, an important fact is that there is a magnetic phase transition in 1L NiPS₃ whose signature in longitudinal transport experiments is qualitatively the same as that of multilayer systems. Remarkably, this kink feature amounts to ~ 15 % of dG/dT (Fig. 4b). This significant result should be emphasized in the main text. Currently, it is almost overlooked. How come should we conclude that this qualitatively the same kink feature in the dG/dT vs T of 1L NiPS₃ comes from the BKT transition whereas the kink for 6L or 2L NiPS₃ comes from the ordinary magnetic phase transition accompanying long-range order? We should not. The conclusion should be that they observed qualitatively the same behavior in dG/dT measurement from 6L all the way down to 1L samples, i.e., the presence of magnetic phase transitions in 1L NiPS₃. The paper should not arrive at a hasty conclusion that there is no long-range order in 1L NiPS₃.*

Reply 8: Conductance vs temperature curves of few-layer NiPS₃ always feature a pronounced kink, which appears as maximum on the derivative dG/dT. However, in our study we observed that the exact shape of G vs T curve is sensitive to the biasing conditions of our FET devices and also varies device-to-device. In particular, we observe variations in the peak shape of dG/dT as well as position of its maximum, which is reflected as error bars on Fig. 1d (see also Fig. 1c and Fig. S2).

We therefore decided to only report the maximum position of dG/dT in our study, which is significantly lower for new 1L devices, (T=124 K) as compared to thicker NiPS₃ (T=140-150K).

Comment 9 *For the above reasons, we cannot recommend publishing this paper in Nature Communications in its current form.*

The authors are advised to make the modifications described so far and change the title and the abstract properly so that the paper does not mislead the readers.

Reply 9: Following the feedback from the reviewer, we have performed new experiments and deeply revised the manuscript. We believe that the updated version is suitable for publication in Nature Communications.

Comment 10: *The following are other important issues to be addressed by the authors to improve the manuscript.*

1. β_{0T} is the angle between the direction of the applied magnetic field and the direction of the Néel vector. The authors should specify the direction of the applied magnetic field for each MR measurement in each figure panel. The readers know that there are uncertainties in the Néel vector direction. Still, at least the other part, the magnetic field direction, is an independent variable that can be set in each experiment. This direction should be specified as an angle value (and as an arrow when appropriate in each panel) in all the figures in the main manuscript and the supplementary information. Also, analyse the error in the direction of the in-plane magnetic field of the experimental setup. How large is the uncertainty in the magnetic field's direction (in-plane angle)? This uncertainty will determine part of the error in β_{0T} . (The other part comes from the uncertainty and inhomogeneity of the Néel vector direction.) Please explain the experimental setup to support the authors' estimation. The readers should know very clearly the sources of the error for β_{0T} and how large each of them is.

Reply 10: Our sample holder is mounted on the probe stick of the cryostat in such a way that the magnetic field is applied entirely in-plane. Since our experimental set-up does not allow for in-situ rotation under an applied magnetic field, we have aligned our samples manually prior to each measurement using an optical microscope. We estimate the misalignment error to be less than 5° . The fitted alignment error obtained from our AMR model is 3.4° as shown in **Fig. R1**, which supports the high precision of our experimental alignment.

We have updated our experimental procedure and provide an estimate for the misalignment error in the methods section of the manuscript.

Comment 11 : *Please show experimental results for an extensive scan of the β_{0T} value at grid points, including the results for large β_{0T} values; for example, in steps of 5 degrees, for 6L, 2L, and 1L samples. Note that the magnetic field direction is not fixed by the device fabrication, such as electrodes, and can be changed in the experiment. Then, present numerical simulations and the best fit for the experimental data using a single AMR value for each sample. This comprehensive comparison between the experimental results and the simulation is required to assess the validity of the AMR theory used in the simulation. Currently, the readers might think that the paper intentionally cherry-picks a tiny portion of the available phase space of the parameters for the MR measurement.*

3. A crucial thing in analyzing the MR data using the AMR model is to fix the AMR overall constant and α_{0T} for a given sample and to fix β_{0T} for one magnetic field direction for this sample and calculate β_{0T} at other magnetic field directions using these parameters because the direction of the magnetic field is tunable and is known; that is, the authors should not fit β_{0T} at each given magnetic field direction. Suppose that they fit $\beta_{0T} = (\text{angle of the magnetic field}) - (\text{angle of } L)$.

Then, one obtained (angle of L) from fitting. Using this parameter, one can calculate β_{0T} = (angle of the magnetic field) – (angle of L) at magnetic fields pointing along different directions.

4. In all samples (6L, 2L, 1L) and at all field strengths, the MR is always negative. However, their interpretation does not prevent positive MR. In fact, the two are equally observable according to Eqs. (1) and (2) of the main manuscript. To make people believe their AMR interpretation, showing results with positive MR is essential. The important parameters that determine the sign of the MR are the angles between the magnetic field, the Néel vector, and the current direction. The angle between the Néel vector and the in-plane magnetic field direction can be changed easily, and the angle between the Néel vector and the current direction is determined by the device setup. They should provide the results for 6L, 2L, and 1L samples with positive MR.

Reply 11: We would like to thank the reviewer for the valuable comments. As we explained earlier, our current experimental set-up unfortunately does not allow us to perform extensive scans with continuous magnetic field angles as grid points. In addition, we intend to present the AMR analysis of MR of NiPS₃ in a separate report.

We agree that the positive MR is not excluded in our experiments. We show an example of positive MR observed in 6L NiPS₃ (**Fig. R2a** and **R2b**) under the magnetic field with an azimuthal angle of +30° with respect to the easy axis. Our AMR model fully captures the shape and the sign of experimental MR curves at fixed simulation parameters while varying only β_{0T} (**Fig. R2c**), which supports the validity of our AMR model in explaining the MR of 6L NiPS₃.

REDACTED

Comment 12: 5. Please provide similar transport (dG/dT vs T) and the MR measurements for a bulk (or a much thicker) sample; these additional data would be very helpful in understanding the trend of how dG/dT and the MR change from bulk all the way down to 1L NiPS₃.

Reply 12: In response to reviewer's comment, we have prepared another NiPS₃ FET from thicker 13 L flake and performed the transport (dG/dT vs T) and MR measurements. These results are now presented together with other thicknesses in **Fig. 1c** and **Fig. 3d**, respectively. We have found that the kink in dG/dT is similar to that observed in the 6L samples. The uniform easy-axis orientation was identified using low-temperature photoluminescence spectroscopy (see **Fig. S6**). When a magnetic field is applied along the easy-axis, MR at 40 K is negative with an amplitude similar to other samples (less than 1%) and the spin-flop field is the same as in the 6L and 2L devices.

Comment 13 : 6. The authors cite Ref. 22 as follows: "Consistently with these considerations, recent Monte Carlo simulations of magnetism in 1L NiPS₃ confirm the occurrence of a BKT phase transition with $T_{BKT} = 141$ K." However, the model considered in Ref. 22 assumes zero in-plane anisotropy, which is not a valid assumption according to the advanced first-principles calculations reported in Ref. 21. Such a model without in-plane anisotropy, in disagreement with first-principles calculations, will of course lead to a BKT transition. Note that first-principles studies like Ref. 21 have quantitatively reproduced the magnetic anisotropy of van der Waals magnetic materials: for example, Xu et al., "Interplay between Kitaev interaction and single ion anisotropy in ferromagnetic CrI₃ and CrGeTe₃ monolayers," *npj Comput. Mater.* 4, 57 (2018), have reported results that are in quantitative agreement with the tiny magnetic anisotropy of 2D CrGeTe₃ [*Nature* 546, 265 (2017)] and large easy-axis anisotropy (along out-of-plane direction) of 2D CrI₃ [*Nature* 546, 270 (2017)] measured from experiments. Therefore, the authors should explain these details in citing Refs. 21 and 22 and cite Xu et al. 2018 *npj Comput. Mater.* paper together with the two back-to-back 2017 *Nature* papers in this context.

Reply 13: We agree with the reviewer and have modified our manuscript to refer to the results of the first-principles calculations of NiPS₃ with in-plane anisotropy, which is now reference 16:

16. Kim, T. Y. & Park, C.-H. Magnetic Anisotropy and Magnetic Ordering of Transition-Metal Phosphorus Trisulfides. *Nano Lett.* 21, 10114–10121 (2021).

We have deeply revised our discussion, which now also demonstrates that our experimental observations of 1L NiPS₃ magnetism are fully in line with the renormalization group calculations of a 2D XY model with crystal-field perturbations, references:

36. José, J. V., Kadanoff, L. P., Kirkpatrick, S. & Nelson, D. R. Renormalization, vortices, and symmetry-breaking perturbations in the two-dimensional planar model. *Phys. Rev. B* 16, 1217–1241 (1977).

15. Seifert, U. F. P., Ye, M. & Balents, L. Ultrafast optical excitation of magnetic dynamics in van der Waals magnets: Coherent magnons and BKT dynamics in NiPS₃. *Phys. Rev. B* **105**, 155138 (2022).

Comment 14 : 7. *The authors mention that the absence of the in-plane easy-axis anisotropy can be explained by the structural symmetry of 1L NiPS₃. This is not true because even if the underlying structure retains the trigonal symmetry, the zigzag magnetic pattern, which is known to be the lowest-energy configuration of the isotropic J_1 - J_2 - J_3 model of NiPS₃, breaks the (trigonal) symmetry of the charge and spin densities and in turn induces an easy-axis anisotropy through spin-orbit coupling. Therefore, the authors should explain in the main manuscript that the widely-accepted, lowest-energy magnetic configuration for NiPS₃ does NOT allow the lack of in-plane easy-axis anisotropy in 1L NiPS₃.*

Reply 14: In the new 1L NiPS₃, we indeed observe MR that reflects hexagonal magnetic anisotropy in 1L NiPS₃ below 60 K.

Comment 15: 8. *Because the system is not a 1D material which can be thought of as serially connected domains of different resistances but a 2D material which is effectively a 2D network of resistances, a simple average of the MR in three different domains, Eq. (S10), which works for a 1D system, will make an additional error from the total MR of the mosaic sample composed of patches of three different domains. Please mention the limitation of Eq. (S10) in the manuscript.*

Reply 15: The previously described multi-domain model has been removed in the revised version of the manuscript.

Comment 16: 9. *In all figures showing the MR results in both the main manuscript and in the supplementary information, show the results for the magnetic field from -14 T to $+14$ T, instead of from -12 T to $+12$ T as done in some figures. Also, for example, in Figs. S13 and S14, provide $B < 0$ regime as well up to -14 T, as done for other cases.*

Reply 16: Most of the experimental data were acquired with the cryostat capable of applying magnetic fields up to ± 12 T. To be consistent, we therefore present all our data over the full range from -12 T to $+12$ T, both in the main text and in the supplementary information.

Comment 17: 10. *In all figures showing simulated MR results in both the main manuscript and in the supplementary information, specify the α_{0T} , β_0 , and the AMR values used in the simulations.*

Reply 17: We have removed the simulated MR curves from the manuscript.

Comment 18: 11. Why does the noise in MR increase if the temperature decreases? This trend is observed in all 6L, 2L, and 1L samples. Please add an explanation of this phenomenon to the main manuscript.

Reply 18: We believe that the phenomenon the review is pointing out could be related to increasing non-linearity of IV curves of our FET devices at low temperature, however we believe that the detailed analysis falls beyond the scope of the present manuscript.

Comment 19: 12. To Fig. 4b, please also add the results for 6L. Moreover, raw experimental data for G and dG/dT should be provided. If there is a space problem, supplementary information is a good option. Currently, only G/G 200K is presented.

Reply 19: We have provided the raw experimental data for G and dG/dT for 6L in **Fig 1c**.

Comment 20: 13. Please specify V_{BG} and V_{SD} in each panel of the figure or the caption for all the figures in the main manuscript and the supplementary information. Readers should easily understand the experimental setup for the results shown in each figure without reading the relevant part of the manuscript.

Reply 20: We have included the corresponding information in the revised version of the manuscript.

Comment 21:

14. In line 207, “which was taken as evidence for the absence of magnetic order” should read “which was suggested as the origin of the absence of a particular peak splitting in the Raman spectrum even if the ground state is magnetic” because Ref. 21 did not claim the absence of magnetic phase transition in 1L NiPS₃; indeed, the opposite was supported in Ref. 21, i.e., the presence of magnetization in 1L NiPS₃.

15. In Fig. S10, the best AMR value for 2L NiPS₃ is -0.37% , but in line 186 of the main text, it is said to be -0.39% . Is this a proper comparison? If so, which value is correct?

16. After Eq. (3) of the main manuscript, please explain the relation between K and D

17. The following sentence between Eq. (S3) and Eq. (S4) is not correct. Please revise it: “ L and M satisfy $L \cdot J = 0$.”

18. In line 183, ‘edges #1 and #2’ should read ‘edges #2 and #3,’ and in lines 185 and 189, ‘edge #3’ should read ‘edge #1.’ Please check these things.

19. In the reference list of the main text, Ref. 8 is the same as Ref. 21. Please delete one of them.

Reply 21: We have corrected the aforementioned inconsistencies and included the corresponding clarifications in the revised version of the manuscript.

Reply to Reviewer #2 :

Comment 1: *Cheon et al. present findings from electrical transport measurements conducted on few-layer NiPS₃ FET devices. They identify a long-range antiferromagnetic (AFM) state in bi- and thicker-layer NiPS₃ by observing the spin-flop transition through the characterization of magnetoresistance in relation to the in-plane magnetic field. In contrast, they do not detect spin-flop-induced resistance changes in monolayer devices, leading them to assert that NiPS₃ represents the first experimental realization of a 2D XY antiferromagnet experiencing the Berezinskii-Kosterlitz-Thouless transition. While the experimental results are intriguing, questions emerge regarding whether these findings adequately support the authors' conclusion and the novelty of their work for publication in Nature Communications.*

1. The authors claim that monolayer NiPS₃ undergoes a BKT transition, based on the observation that no abrupt change in magnetoresistance is observed above 10 T. However, it is important to note that there are several other possible explanations for this experimental observation, which the authors do not explicitly rule out. For instance, recent studies have shown that NiPS₃ transitions from a three-dimensional antiferromagnetic state to a two-dimensional "vestigial nematic" state due to strong quantum fluctuations as its thickness decreases. In this scenario, the spins are randomly canted away from the zigzag chain direction, leading to a loss of long-distance spin coherence. The authors must present additional evidence to establish a robust connection between their experimental findings and the proposed conclusion.

Reply 1: We thank the reviewer for the positive assessment of our work.

Recent Raman spectroscopy and linear dichroism studies indeed report a vestigial nematic state in few-layer NiPS₃ assigned to the partial melting of the conventional zigzag AFM order due to enhanced fluctuations [Nano Lett. 2024, 24, 24, 7166, Nat. Phys. 2024, 20, 1764]. On the other hand, we observe virtually the same spin-flop transition from bulk down to 2L NiPS₃ at the critical magnetic field of approx. 10.5 T below 100 K (Fig. 3d), which is identical to what has been reported in bulk NiPS₃ by other groups [Phys. Rev. B **108**, 115149 & *Nat Commun* **15**, 7841 (2024)]. We therefore conclude that the suggested partial melting of the AFM order into the vestigial nematic state does not significantly alter the metamagnetic transitions and hence our interpretation of magnetism in the few-layer NiPS₃.

In the updated manuscript, we have provided new experimental results of 1L NiPS₃. The key finding is the observation of MR and spin-flop transition at the critical field around 5T when the magnetic field is swept along its high symmetry of crystal orientations (three directions 120 ° apart from each other). This finding allows us to determine the existence of long-range magnetic ordering in 1L below 60 K and reveals the presence of hexagonal magnetic anisotropy that governs

its magnetic ground state. The aforementioned Raman studies of few-layer NiPS₃ fail to probe the magnetism in monolayer NiPS₃.

Comment 2: *Several groups have shown that the BKT transition may occur in monolayer NiPS₃ (e.g., 10.1103/PhysRevB.107.L220407) using experimental techniques like helicity-resolved Raman and ultrafast spectroscopy. In this paper, the authors employ a different approach (electrical transport) to reach a similar conclusion, which diminishes its novelty and significance.*

Reply 2: The reviewer is right that our work takes the different experimental approach based on magnetotransport measurements to probe the magnetic order of an intralayer antiferromagnet down to the monolayer limit. However, we disagree with the reviewer that the previous reports about few-layer NiPS₃ diminish the novelty of our work, in particular:

i) no experimental report probes the spin-flop transitions in few-layer NiPS₃, whereas we report such transitions down to monolayer thickness

ii) Raman or linear dichroism studies fail to probe the magnetism in monolayer thickness, whereas we map the phase diagram of 1L NiPS₃. The updated version of the manuscript reports two distinct magnetic phase transitions, which we assign to BKT transition (~ 124 K) and onset of long-range zigzag magnetic ordering (~ 60 K). These results are consistent with the renormalization theory with crystal-field perturbations, which predicts that the 2D XY system with hexagonal symmetry will undergo two magnetic phase transition at T_{BKT} and T_{C} , with $T_{\text{BKT}} > T_{\text{C}}$. In the temperature range $T_{\text{C}} < T < T_{\text{BKT}}$, the system lacks long-range order, while below T_{C} the system establishes the magnetic order with one of the six degenerate ground states. We believe that our experimental observations have not been reported before in monolayer NiPS₃ and broadly in any other intralayer 2D antiferromagnet.

Lastly, we would like to elaborate on the study mentioned by the reviewer [Phys. Rev. B 107, L220407 (2023)]. They report a transition temperature for 1L NiPS₃ of 140 K, which is much higher than our observations (124 K). More importantly, their conclusions largely rely on analysis of extremely broad two-magnon scattering background of Raman spectra, which is present even at room temperature (paramagnetic state) and “does not guarantee the existence of magnetic order”, as pointed out by the authors of the study themselves. The contribution of the two-magnon scattering in the Raman spectrum of monolayer NiPS₃ was also studied by another group [Nat. Commun. 2019, 10, 345], who concluded that “the two-magnon feature is not well defined” in monolayer NiPS₃ and finally came to conclusion that “antiferromagnetic ordering is significantly suppressed in monolayer NiPS₃”. We therefore believe that more work is needed to probe the two-magnon scattering in NiPS₃ and reveal its connection to the underlying magnetic order. In contrast, our measurements of conductance vs temperature provide direct and unambiguous evidence of phase transition at 124 K, which we reproduced among multiple samples.

Comment 3: *The manuscript is titled “Nature of 2D XY Antiferromagnetism in a Van der Waals Monolayer.” However, much of the main text centers on the experimental results from a 6-layer device. A different title that more accurately reflects the content of the manuscript might be more appropriate.*

Reply 3: We agree with the reviewer that the previous version of the manuscript largely focused on 6L NiPS₃. In the updated version we removed the discussion about anisotropic magnetoresistance of 6L NiPS₃ and shifted the emphasis to experimental data and discussion of 1L NiPS₃, ultimately mapping the phase diagram of 2D XY antiferromagnet. Therefore, we believe that our title now more accurately reflects the content of our manuscript.

Comment 4: *According to the text, conducting polarization-dependent PL measurements on a monolayer device is challenging. How do the authors determine the crystal direction? Furthermore, the authors state that there are three types of magnetic domains; what are the sizes of these domains, and how are they distributed?*

Reply 4: Indeed, due to challenges associated with the polarization-dependent PL measurements in few-layer NiPS₃, we only used them to determine the orientation of the Néel vector for samples > 3L. For thinner samples we relied on the edges of the exfoliated crystal. Indeed, it has been demonstrated that the family of metal thiophosphates exhibits a strong tendency for exfoliated crystals to terminate with 120° edges, corresponding to either zigzag or armchair orientations [*ACS Nano* 2016, 10, 9, 8980–8988]. This characteristic is also observed in our exfoliated 6L and 13L samples. The polarization-resolved PL measurements on these samples reveal that the Néel vector lies along one of the crystal edges. We therefore conclude that the edges of NiPS₃ crystals are oriented along the zigzag direction. We also find that the 2L and 1L NiPS₃ crystals show 120°-terminated edges and indeed observe spin-flop transitions when magnetic field is oriented along them.

We have presented new 1L data where we observe MR below 60 K. The MR response is symmetric in magnetic field taken 120° apart and exhibits behavior that is not seen in thicker layers and cannot be explained by a system with uniaxial magnetic anisotropy. We interpret this as evidence of the formation of magnetically ordered domains, each possessing an easy axis (Néel vector) aligned along one of the three possible zigzag orientations. Unfortunately, the size and distribution of the magnetic domains cannot be determined directly from our measurements. However, if we assume that domain averaging accounts for the absence of Raman and [*Nat Commun* 10, 345 (2019)] linear dichroism [*Nano Lett.* 2024, 24, 24, 7166–7172] signals in 1L NiPS₃, then the domain size should be smaller than the optical laser spot area (i.e., less than 1 μm²). We estimate an approximately equal distribution of domain orientations, as we observe similar MR amplitudes across different magnetic field directions.

Comment 5: 5. Typically, monolayer van der Waals materials are less stable than multilayer samples. Sample degradation could lead to changes in electrical and magnetic properties. Did the authors assess the stability of the device?

Reply 5: Indeed, monolayer van der Waals materials are often prone to degradation in ambient conditions. However, unlike e.g. CrI₃, NiPS₃ is more stable and hence many studies handle it in air [Applied Surface Science, 504, 144405]. We never exposed the unencapsulated samples to ambient conditions and handled the samples only in nitrogen-filled gloveboxes. Our electrical data from the encapsulated 1L FETs show that the transfer curves of 1L remain identical after several warm-up / cooldown cycles, in between which the encapsulated devices were in ambient atmospheric conditions for several hours (**Fig. R3**). The only difference we observe is the small shift in the threshold voltage of a transistor, which is likely due to excess electrical charges on the surface of the van der Waals heterostructure due to ambient adsorbates.

Figure R3. Stability of 1L device. **a.** Transfer curves measured at 40 K after multiple cooldowns. Before each cooldown, the sample was exposed to ambient atmospheric conditions for few hours. **b.** Source-drain current (I_{SD}) vs T during 1st (left panel) and 2nd (right panel) cooldown. We observe the same critical temperature for the magnetic phase transition.

Comment 6: Due to these reasons, I cannot recommend publication of this work in Nature Communications. It may be suitable for other specialized journal after major revisions.

Reply 6: We believe that the experimental findings in 1L NiPS₃ in our revised manuscript strongly supports our claim and thus meets the high standard of publishing in Nature Communications.

Response to reviewers' critique for the manuscript "*Nature of 2D XY antiferromagnetism in a van der Waals monolayer*" by Cheol-Yeon Cheon, Volodymyr Multian, Kenji Watanabe, Takashi Taniguchi, Alberto F. Morpurgo, and Dmitry Lebedev.

Reply to Reviewer #1:

We have carefully read and considered all the comments of reviewer 1, however we find that the vast majority of them are unmotivated and inappropriate. Before turning into point-by-point response, let us provide some general comments:

- 1) Most comments by the reviewer do not pinpoint a precise error or inconsistency in what we did or what we wrote in the manuscript. The comments seem more to indicate that the reviewers have their own pre-made scenario, and they are requesting us to provide data and discussion which will fit their expectations.
- 2) The style of the review is inappropriate: instead of criticizing a particular interpretation or statement in our manuscript, the reviewer provides pages of statements, which are unclear and not always internally consistent. This makes it impossible to grasp what exactly is questioned in our results and interpretations and why, and how we are supposed to react.
- 3) The requests of the reviewer for further measurements that we should do (or explanation we should give) are not motivated by any specific reason. Experiments for the sake of experiments.
- 4) The extreme level of details in the comments and in the requests of the reviewer, in conjunction with them not pointing out errors or inconsistencies in our work, gives a strong impression that they are trying to stall or delay publication rather than giving a constructive assessment of our results.
- 5) Last but not least, instead of providing a scientific critique and feedback, the reviewer tends to supervise the work, e.g. outline figures we need to provide (including axis dimensions, color schemes, etc). We find such behavior inappropriate, unprofessional, and disrespectful.

Comment 1

Based on the criticisms and comments of the s, the authors have significantly revised the manuscript. In particular, they said that what they had thought to be a monolayer (1L) sample was actually a bilayer (2L) and that they had found true 1L samples. Moreover, they abandoned their theoretical model on the spin-flop transition because the model does not work very well for 1L and 2L samples.

The authors have also entirely changed their central claim. Now they say that there is indeed a magnetic phase transition in their 1L sample, albeit at a lower temperature ($T^ \approx 60$ K). They bring an old paper [Ref. 42, which is J. V. Jos'e et al., Phys. Rev. B 16, 1217 (1977)] to explain that there are two phase transitions: a BKT transition at $T_c = 124$ K and a normal magnetic phase transition at $T^* = 60$ K, below which a long-range order is formed.*

We appreciate the authors for performing additional experiments and trying to clarify the issues. However, for the following reasons, we find that the new, completely different manuscript has its own serious problems. We still believe that the subject is of importance to the general readership of Nature Communications; however, the conclusion is not supported by the experimental data, as we explain in detail. Therefore, we cannot recommend the publication of this paper in Nature Communications in its current form. However, we think that the authors may be given one last chance to revise the paper so that they reach a conclusion consistent with their experimental data and provide their high-quality experimental data and possible explanations.

1. *First of all, the authors provide no experimental or theoretical evidence to support their claim that what they observed for the magnetoresistance (MR) of 1L NiPS₃ is what was proposed by the Jos'e et al.'s paper other than the fact that, according to the authors' claim, only 1L NiPS₃ shows some two-step-like temperature dependence in the measured MR, although we also doubt this claim about 1L NiPS₃ as we detail later. They do not compare the detailed contents of the paper by Jos'e et al. or any other theoretical paper with their experimental data (conductance at $B = 0$ and MR). Importantly, the paper by Jos'e et al. is on the XY model with hexagonal symmetry-breaking term at $B = 0$; however, the measured conductance at $B = 0$ as a function of temperature does not show the two-temperature feature that could possibly be deduced from the model. (We can only say this weakly because the paper by Jos'e et al. did not make any predictions on the conductance even for the $B = 0$ case.) Still, the authors apply the paper by Jos'e et al., which is on the $B = 0$ case only, to their MR data, i.e., experimental data performed under a non-zero magnetic field. Obviously, this inconsistent, cherry-picking comparison is groundless. Moreover, their experimental data for $B = 0$ can be explained quantitatively by first-principles calculations assuming $B = 0$, as we explain in our next point. However, we do not blame the authors for this complete absence of evidence because this subject actually belongs to uncharted territory, as we also explain in our next point.*

Reply 1

In this work, we report multiple experimental indications of a phase transition at $T^=60$ K in 1L NiPS₃, including the onset of magnetoresistance (MR), a spin-flop transition, and enhanced current fluctuations. Notably, the current fluctuations are present at $B=0$ T (see Fig. S8) and occur only in 1L NiPS₃, not in thicker layers.*

The $B=0$ T conductance behaviors in 1L NiPS₃ — distinct from other thicknesses — cannot be explained by the result of first-principles calculations based on $B=0$ T [Ref. 19, Nano Lett. 21, 10114 (2021)]. What the reviewer states —namely that our results are incompatible with the theory for the $B=0$ T case, i.e., the paper by Jos'e et al. —is incorrect: from the conductance data

at $B=0T$ alone, we determine two characteristic temperatures in 1L NiPS₃, which is precisely the conclusion expected from the theory of José *et al.*

The work by José *et al.* is a landmark theory paper and is regarded as a milestone in the research on 2D XY models. This work is a renormalization group study of the 2D XY model in presence of anisotropy term, with all possible magnetic anisotropy terms allowed by crystalline symmetry (2-,3-,4-, and 6-folds). The result of the study is simple and clear: only in the presence of a 6-fold anisotropy two phase transitions at different temperatures can be seen in the XY model. For all other anisotropy terms, only one critical temperature is present. What we find experimentally – evidence for two phase transitions only in 1L NiPS₃ and not in thicker layers – is precisely what is expected by José *et al.*

While renormalization group studies do not give predictions for details of the temperature and magnetic field dependence of MR, we use MR (and current fluctuation) measurements to detect phase transitions. We are not “cherry-picking” the theory: currently, to the best of our knowledge, there are no predictions of magnetoresistance for the XY model under finite magnetic field, and we hope our experimental work will motivate such theoretical studies.

In addition to the 2D XY model with anisotropy term considered by José *et al.*, one could also consider so-called p -state clock models, where instead of continuous spins rotation the spins are restricted to a discrete angles in plane:

$$H = -J \sum_{\langle ij \rangle} \vec{s}_i \cdot \vec{s}_j = -J \sum_{\langle ij \rangle} \cos(\theta_i - \theta_j)$$

where $\theta = 2n\pi / p$, with $n = 1, 2, \dots, p$. These models are widely used in the context of 2D magnets. Studies show that for $2 \leq p \leq 5$, the system undergoes one phase transition into ordered phase, whereas for $5 \leq p \leq \infty$ two phase transitions are expected, according to the following references:

i) G. Ortiz *et al.*, Dualities and the phase diagram of the p -clock model, Nuclear Physics B 2012 854 780

ii) P.P Orth, Emergent criticality and Friedan scaling in a two-dimensional frustrated Heisenberg antiferromagnet, PRB 2014 89, 094417

iii) Yutaka Okabe and Hiromi Otsuka BKT transitions of the XY and six-state clock models on the various two-dimensional lattices J. Phys. A: Math. Theor. 2025 58 065003

Therefore, both renormalization group analysis of the 2D XY model with hexagonal anisotropy and six-state clock models predict two phase transitions, augmenting our experimental observations.

We have added discussion of the six-state clock models to our manuscript.

Comment 2

2. *Jos'e et al.'s paper assumes a simple hexagonal symmetry-breaking term of the form $\cos(6\theta)$ in addition to the perfect XY Hamiltonian. However, this model does NOT reflect the nature of the effective Hamiltonian for NiPS₃. Indeed, according to first-principles calculations [Ref. 19, which is Nano Lett. 21, 10114 (2021)], the ground state of NiPS₃ is six-fold degenerate, but for a different reason: three degenerate zigzag chain directions separated by an angle of $2\pi/3$ for the antiferromagnetic spin configuration and two options for the collinear spin alignment along the given zigzag chain direction (the two ground-state options are connected by inverting all the spins). However, this 6-fold degeneracy of NiPS₃ is qualitatively different from the simple, naive $\cos(6\theta)$ term of Ref. 42. According to the state-of-the-art first-principles calculation in Ref. 19, this 1L NiPS₃ (hosting 6-fold-degenerate ground states) undergoes a magnetic phase transition, and the calculated T_c drop of 3 K in going from 3L to 2L and that of 15 K in going from 2L to 1L agree perfectly with the measured values of 3 K and 16 K, respectively. Therefore, the results on the conductance at B = 0 could indeed be strong quantitative evidence of qualitatively the same magnetic phase transition in NiPS₃ from bulk all the way down to 1L samples in agreement with state-of-the-art first-principles calculations assuming zero magnetic field. This is a beautiful experimental verification of the prediction from state-of-the-art first-principles calculations. To our knowledge, we have not seen a plot like Fig. 1d in any previous studies. This excellent agreement between theory and experiment should be mentioned and emphasized in the manuscript.*

Reply 2

We disagree with the reviewer that a $\cos(6\theta)$ term is too simplified to describe real NiPS₃ monolayers, and that is why the predictions of the paper of José et al do not necessarily apply. Renormalization group relies on generic symmetry properties of the terms in the Hamiltonian not on their precise details, because it is the symmetry properties that determine the stable phases and behavior of a system near phase transitions.

The reviewer also mentions that a $\cos(6\theta)$ term in the Hamiltonian is different from the degeneracy of the ground states. Of course it is different: when discussing phase transitions the Hamiltonian and the ground states generically have different symmetries because of spontaneous symmetry breaking. We find it meaningless to state that a term in the Hamiltonian is different from a ground state degeneracy.

Comment 3

3. *If the experimental results provide evidence for the existence of the magnetic, ordered phase in NiPS₃ at B = 0 from bulk all the way down to monolayer samples, then the natural question arising would be how to explain the experimental results with a non-zero in-plane magnetic field from bulk all the way down to monolayer NiPS₃. So far, nobody knows about the temperature-dependent MR and spin-flop transitions in 1L NiPS₃, which hosts such complicated 6-fold degenerate ground states (qualitatively different from the simple model in Ref. 42, unsuitable for 1L NiPS₃). However, this subject is far beyond the scope of this article reporting on an experimental study. This paper beautifully provides experimental evidence for the magnetic phase transition in 1L NiPS₃ at B = 0, qualitatively the same as in thicker samples; If published in Nature Communications, this paper will lead to theoretical studies on this important subject, the MR and spin-flop transitions of 1L NiPS₃, which hosts the exotic 6-fold degenerate ground states based on the 3-state nematicity, under an in-plane magnetic field.*

Reply 3

We agree with the reviewer that our experimental findings will promote theoretical studies of the 6-fold degenerate ground state in 1L NiPS₃.

Comment 4

4. As we explained before, the significant drop in T_c in going from 2L to 1L NiPS₃ ($T_{2L} - T_{1L} = 16$ K) in comparison with the drop in T_c in going from 3L to 2L ($T_{3L} - T_{2L} = 3$ K) is a quantitative difference, not a qualitative one. This quantitative difference is due to the strong interlayer spin exchange interactions and is unique for NiPS₃. Other compounds like MnPS₃ or FePS₃ do not show such a significant change in T_c in going from 2L to 1L because their interlayer exchange interactions are much weaker than those of NiPS₃, as revealed and explained by first-principles calculations in Ref. 19. The authors should discuss the effect of unusually strong interlayer exchange interactions in NiPS₃ in the main manuscript.

Reply 4

In the previously submitted version of manuscript, we actually did discuss the effect of interlayer exchange interactions of NiPS₃ for the distinct magnetic phase diagram of 1L and 2L NiPS₃. Therefore, it is unclear to us what the reviewer is exactly requesting.

The reviewer emphasizes at length that the strength of interlayer interaction is stronger in NiPS₃ than the compounds such as MnPS₃ and FePS₃. However, this comparison is not relevant to the focus of our work, because it is well established, based on direct experimental measurements, that both MnPS₃ and FePS₃ exhibit out-of-plane easy-axis magnetic anisotropy. Regardless of the relative strength of their interlayer interactions, the magnetic anisotropy in MnPS₃ and FePS₃ is inherently incompatible with XY-model physics. Why the reviewer fails to mention the well-known (and key) difference in the magnetic anisotropy of NiPS₃ and MnPS₃ and FePS₃ is totally unclear to us.

On the contrary, in NiPS₃, the interlayer interaction is critical because its presence would unavoidably promote 3D spin correlation, masking the intrinsic 2D XY behavior in NiPS₃. That is precisely what we observed experimentally, and we stated “...truly 2D limit of magnetism in NiPS₃ is only reached when the thickness is reduced to an individual monolayer.” in our submitted version of manuscript.

Comment 5

In this way, we may be able to understand many of the experimental results that were assigned to two-step magnetic phase transitions. (i) For example, the 1L data obtained at $T = 40$ K presented in Fig. 3d shows a sharp decrease in MR at $B \sim 5$ T. In contrast, the MR of 1L NiPS₃ at $T = 5$ K shown in Fig. 4a drops sharply around $B = 7$ T and resembles the 2L data at $T = 40$ K in Fig. 3d. A closer examination of the 2D scan of the MR of 1L NiPS₃ shown in Fig. 4c reveals that the magnetic field strength at the onset of the sharp decrease in MR reaches a minimum around ~ 33 K, which is close to the temperature ~ 40 K at which the 1L data in Fig. 3d was obtained. In this context, we can understand some of the differences between 1L and 2L MR data as being quantitative, rather than qualitative. Please revise the manuscript in light of this observation. (ii) Also, it is clear that in going from 6L to 2L, B_{sf} (spin-flop field) at $T = 90$ K decreases from 11.5 T (Fig. 2e or Fig. S4b) to 10.7 T (Fig. S7c).

Reply 5

In this comment, it is unclear to us what conclusions we should derive from comparing data between different thicknesses at different temperatures (e.g. 1L at 5K vs 2L at 40K). It is also

unclear to us what should be revised and why. Unfortunately, the reviewer fails to provide an explanation.

Comment 6

How these trends evolve in samples of different thicknesses could provide some critical clues and additional experimental data that are valuable for future studies. Currently, the authors provide a detailed analysis of 6L and 1L samples only. They do not provide any 2D scan for 3L, and, in the case of 2L sample, they provide the 2D scan only in a very narrow temperature range (Fig. S7b). The readers should be able to view the complete experimental data and know the trend on how the 2D scan varies from 13L to 6L, to 3L, to 2L, and to 1L NiPS3. We request the authors to provide the following data in the Supplementary Information.

(1) *Provide complete 2D scans of magnetoresistance (MR) vs temperature and magnetic field for T from 0 K to 180 K and B from -12 T to $+12$ T (see panel a of Fig. 1 of this report) for all the samples: 13L, 6L, 3L, 2L, 2L (, 2L, 2L) (they have three or five different 2L samples), 1L, 1L, 1L (the authors also have three different 1L samples as shown in Fig. S2). Also, we suggest that the authors provide more 6L, 5L, 4L, 3L, and, most importantly, more 1L samples and provide the same 2D scans. These additional samples will be very important for the readers to distinguish which features are universal and which features depend on the quality of the sample and the device configurations, such as VSD and VBG.*

(2) *For the 2D scans, use the same color bar in all cases (samples of different thicknesses) so that the readers can make an unbiased, fair comparison by themselves.*

(3) *For each 2D scan of MR, add a dR/dB 2D scan as well, as done in Fig. S4a. See panel b of Fig. 1 of this report. Use the same color map in all dR/dB 2D scans, also for samples with different thicknesses.*

(4) *For each of the 2D scans, add a graph showing B_{sf} vs. T (see panel c of Fig. 1 of this report). For each T , there should be two B_{sf} values, one positive and one negative. The range for B_{sf} should also be from -12 T to 12 T, and that for T should be from 0 K to 180 K. We know that there will be no data points at high temperatures; however, for the readers to be able to fairly compare samples of different thicknesses, it is very important that all the graphs (corresponding to different thickness samples) show exactly the same range of B_{sf} and T .*

(5) *Very importantly, for each data point in the graph described in (4), provide the MR vs. B graph (see panel d of Fig. 1 of this report) so that the readers can know how they extracted B_{sf} from the raw MR data. (Also, explain how B_{sf} was chosen in the revised manuscript.) If there are too many curves, they can split the panel into a few panels but should not leave out any curves.*

(6) *Regarding (2) and (3), please provide additional (separate) figure(s) with different color bar(s) if the authors judge that using a single universal color bar for all samples may miss some essential features. However, in this case, the authors should also show all the sample results in the additional figure(s) using the same color bar in each additional figure. So, the authors may make 2 to 3 figures in total: Each figure shows all sample 2D scans with the same color bar. However, the plots for B_{sf} vs. T and the plots for MR vs. B (panels c and d of Fig. 1 of this report, respectively) do not need to be repeated in the additional figure(s).*

(7) *A closer look at Fig. 4b shows that the temperature at which dG/dT becomes negative closely matches T^* ($= 60$ K). Interestingly, if we estimate the temperatures at which dG/dT turns negative in Fig. 1c for various thicknesses, we find that the temperatures are around 80 K and 100 K for 2L and 6L samples, respectively. These temperatures are similar to those below which the sharp features in the MR at $B \sim 8-10$ T appear in each case (see Fig. S7a and*

Fig. 2c, respectively). Is this correlation that appears across multiple datasets in this paper a coincidence? Provide additional data to judge whether the three 1L devices presented in Fig. S2 all show the switching behavior at the same temperature.

(8) In addition, thoroughly investigate how the 2D scans depend on VSD and VBG for samples of different thicknesses, including the 1L, 1L, 1L, 2L, 2L, 2L, (2L, 2L,) and 6L samples.

Reply 6

We find that the aforementioned requests by the reviewer are unmotivated, inappropriate, and unfeasible from an experimental point of view. Let us comment on all these points separately:

1. Unmotivated. The reviewer does not explain why the requested measurements need to be performed. They only provide general statements like “could provide some critical clues”, “experimental data that are valuable for future studies”, and “the readers should be able to view the complete experimental data”. We find such motivation completely unacceptable given the amount of work required to collect, analyze, and plot the requested data.

2. Inappropriate. According to *Nature*'s guidelines for referees, the role of referees is to provide a scientific evaluation of the manuscript. Here the reviewer takes on the role of a supervisor, instructing us how to collect, organize, and plot our data (color bars, axis limits, etc.) On the final page of their response, they even outlined a figure to be included by us with 36 panels! We simply find this level of instruction excessive, unprofessional and disrespectful. It absolutely does not align with the spirit of constructive peer review.

3. Unfeasible. The amount of data requested by the reviewer is enormous and, based on our estimates, will take years to collect. The phase space indicated by reviewer includes the following dimensions: sample thickness, temperature (from 0 K!), magnetic field strength, magnetic field orientation, Vsd, and Vbg: total 7 dimensions.

We are willing to measure, analyze, and plot particular cuts of this 7D space. Unfortunately the reviewer fails to provide an appropriate motivated request.

Comment 7

5. B_{sf} is a rapidly increasing function of temperature. Therefore, we do not know if there will be a spin-flop transition in 1L NiPS₃ at higher temperatures if higher magnetic fields are applied. The authors should explain this temperature dependence of B_{sf} and the possibility of observing spin-flop transitions in 1L NiPS₃ at higher temperatures under higher magnetic fields in the main manuscript.

Reply 7

As shown in **Figure 2e**, the mean-field approximation of spin-flop fields in an uniaxial system, $H_{sf}^2 = \frac{2K}{\chi_{\perp} - \chi_{\parallel}}$, agrees well with our experimental observations in 6L NiPS₃. It is supported by the excellent proportionality between the extracted spin-flop fields and the inverse square root of in-plane magnetic susceptibility anisotropy of bulk NiPS₃. The spin-flop field above T=100 K can be estimated using this relation, which indeed increases with temperature far beyond the limit of our experimental apparatus, B=12 T (see **Figure R1**).

In contrast to 6L, the observed temperature dependence of B_{sf} in 1L clearly indicates that the absence of a spin-flop transition at higher temperatures above T=60 K cannot be attributed to limitations in the accessible magnetic field range.

Figure R1. B_{sf} of 1L and 6L NiPS_3 as a function of temperature. B_{sf} for 6L above $T=100$ K is extrapolated using the relation $H_{sf}^2 = \frac{2}{\chi_{\perp} - \chi_{\parallel}}$, based on the extracted anisotropy energy (K) and the bulk susceptibility data from ref.16. The grey K -shaded region indicates the field range beyond our experimental capabilities. The vertical lines mark the critical temperatures: T_c and T^* (1^{st} and 2^{nd} transition temperature of 1L), and T_N (Néel temperature of 6L).

Comment 8

6. The sentence in the abstract “While bilayer and multilayer NiPS_3 exhibit a single magnetic phase transition into a zig-zag antiferromagnetic state driven by uniaxial anisotropy, monolayer NiPS_3 undergoes two magnetic transitions, with a low-temperature phase governed by in-plane hexagonal magnetic anisotropy” is inconsistent with the data presented in this paper. The temperature below which the spin-flop transition starts to occur and T_c are also different in the cases of 2L, 3L, 6L, and 13L samples. For example, for the 6L sample, a spin-flop transition is observed for B within $-12 \sim 12$ T only up to 100 K (see Fig. 2d and Fig. S4a), which is significantly lower than $T_c \sim 149$ K. Please explain this difference between the relevant temperatures for magnetic phase transitions and spin-flop transitions in samples thicker than 1L NiPS_3 in the abstract and also in other parts of the paper that have similar claims.

Reply 8

We disagree with the reviewer that the quoted statement is inconsistent with the experimental data. For samples $>1\text{L}$ B_{sf} diverges when approaching T_N , as difference between the susceptibilities vanishes (see the equation indicated in the response to the previous comment). We therefore detect B_{sf} up to the limit of our measurement apparatus: 12 T.

Comment 9

7. The sentences “We recall that the magnetic structure of bulk NiPS_3 is governed by a strong easy-plane anisotropy ($D_z \approx 0.2$ meV per Ni), which confines the spins to predominantly lie in the ab -plane, and a weak uniaxial anisotropy ($D_x \approx -0.01$ meV per Ni). The latter is believed to originate from interlayer interactions due to the monoclinic layer stacking, and orients the spins (and hence the Néel vector, L) along the crystallographic a -axis (Figure 1a)”

is not correct. The zigzag anti-ferromagnetic spin configuration can also significantly affect the direction of the spin polarization. The magnetic anisotropy arising from the intralayer interactions can be an order of magnitude larger than that from the interlayer interactions as explained in Ref. 19. The authors should revise the explanation.

Reply 9

It is unclear which sentence does the reviewer view as incorrect. The statements about the type and the strength of magnetic anisotropies are supported by literature data, in particular the anisotropies were explicitly measured using neutron scattering in several experimental papers, such as:

Wildes, A. R. *et al.* Magnetic dynamics of NiPS₃. *Phys. Rev. B* 106, 174422 (2022).

Scheie, A. *et al.* Spin wave Hamiltonian and anomalous scattering in NiPS₃. *Phys. Rev. B* 108, 104402 (2023).

The exact origin of magnetic anisotropies, particularly uniaxial anisotropy, is not fully established in the literature so far. Trigonal distortion of the Ni²⁺ octahedra is known to give rise to uniaxial anisotropy in bulk NiPS₃. Our experimental data show that this anisotropy is not present in 1L, which may be related to structural or symmetry changes in the absence of interlayer coupling, although a direct link between interlayer interactions and trigonal distortion has not yet been reported or studied.

Nevertheless, our experimental data clearly demonstrates that interlayer interactions, are essential ingredient for the occurrence of uniaxial anisotropy. This is supported by our observation that uniaxial anisotropy is absent in 1L NiPS₃, where interlayer interactions are not present.

We modified the text of our manuscript to better reflect our experimental observations and claims.

Comment 10

8. *The 2L data in Fig. 1c and Fig. S2b of the current version and Fig. 4b of the previous version (both the 2L and the 1L data, since the authors now claim that the 1L sample was actually also a 2L one) are very different from each other. Please explain why any currently presented 2L data are different from the previous data (for 2L or 1L - which is supposedly 2L - samples). Please show (1) all 3 curves in Fig. S2b and the 2L data (and also the 1L data because it was actually also a 2L sample) of the previous version in Fig. 1c of the revised manuscript (i.e., five different curves together).*

Reply 10

As requested, we have plotted in **Figure R2** the dG/dT curves from three 1L devices (**panel a**, identical to Fig. S2b) and two 2L devices (**panel b** identical to Fig. 1c; **panels c-d** identical to Fig. 4b from the previous version of the manuscript). The 2L data presented in current manuscript (Fig. 1c) and the previously reported 2L data (Fig. 4b of the previous version) were taken under different bias configurations (See **panel b** and **c** in **Figure R2**).

As previously explained in our rebuttal (see reply #8 of comment #8), the exact shape of G vs T curve is sensitive to the biasing conditions of our FET devices and also varies device-to-device. We note that we have already taken this variability into account when determining the critical temperature as a function of thickness, as reflected in the error bars shown in **Fig. 1d**.

What is essential to highlight is that both curves in panel b and c show a kink around 140 K, indicating a transition temperature that is clearly higher than that observed in 1L. Therefore, the data do not support the reviewer's point that the G vs T data for previously misidentified 2L sample is very different from the data shown in the current version of the manuscript.

Figure R2. a-d. dG/dT vs. T curves for 1L (a) and 2L (b-d) devices. The red and black vertical dotted lines indicate the position of the dG/dT peak for 1L and 2L, respectively.

Comment 11

9. For the 6L sample, the sharp spin-flop transition signatures do NOT exist near 150 K; rather, the signature is visible only below 110 K. Should we interpret the results so that there are also two different phase transitions in 6L NiPS3? We do not think so. As we explained before, the authors should provide the 2D scan as in Fig. 2d and Fig. 4c for all the other samples, 13L, 3L, 2L, etc. Also, provide 2D scans for different samples of the same thickness, i.e., three (or five, including the ones from the previous version of the manuscript) such 2D scans for the 2L samples and three 2D scans for the 1L samples, since the authors already have them. If the authors have more 13L, 6L, and 3L samples, they are advised to show the 2D scans for them as well. The readers should be able to compare these thorough 2D scans before drawing any conclusions.

Reply 11

We find that these requests by the reviewer are unmotivated, inappropriate, and unfeasible from an experimental point of view. See more detailed response above (comment 6)

Comment 12

10. Similarly to the results in Figs. 3b and 3c, what do the results look like if the magnetic field is perpendicular to the edges of 2L and 1L NiPS3 samples? Please provide figures like Figs. 3b and 3c for the case where the magnetic field is perpendicular to the edges for all three (or five) 2L samples and all three 1L samples

Reply 12

Similarly to the previous comment, we find that these requests by the reviewer are unmotivated, inappropriate, and unfeasible from an experimental point of view. See more detailed response above (comment 6)

Reply to Reviewer #2:

Comment 1:

In the revised manuscript, the authors have fabricated and characterized a monolayer NiPS₃ FET, observing distinct transport behavior compared to FETs based on NiPS₃ nanoflakes thicker than two layers. They propose two-step magnetic phase transitions in 1L NiPS₃: a BKT transition near 120 K and a second transition below 60 K leading to long-range magnetic order. While the revised manuscript shows improvement over the previous version, and the experimental findings in 1L NiPS₃ FET are intriguing, several key questions must be addressed before further consideration for publication:

1. The magnetoresistance measurements in Fig. 3C are conducted along three characteristic directions (aligned with hexagonal crystalline axes, 120° apart) to observe spin-flop behaviour. To clarify whether the magnetic symmetry of 1L NiPS₃ is isotropic or hexagonally symmetric, the authors should also examine non-special angles.

Reply 1:

We thank Reviewer #2 for acknowledging the improvements in the manuscript and finding our updated results intriguing.

The observation of spin-flop transition already implies finite magnetic anisotropy in 1L NiPS₃. In the case of isotropic magnetic symmetry, there will be no change in energy when the direction of the magnetic order is rotated within the plane. As a result, Néel vector will always orient perpendicular to the applied in-plane magnetic field and no spin-flop is expected at a finite field for isotropic system (i.e. for the isotropic case, the spin-flop transition would be at $B \sim 0$ T).

Nevertheless, following the suggestion by the reviewer, we performed MR measurements on 1L with the magnetic field applied perpendicular to edge #1 (i.e. between edges #2 and #3, see **Figure 3a**), using the same temperature and bias configuration as in **Figure 3c**. As shown in **Figure S9** below, the MR curve at this field direction exhibits qualitatively different behavior compared to the three directions presented on **Figure 3c** (edge #1, #2 and #3).

In particular, negative MR develops at lower fields and saturates to the same MR (%) as observed for the field applied along other edges. Such behavior closely resembles the results of calculation based on modified spin-flop model that considers multiple symmetry-related magnetic sublattices in hexagonal antiferromagnets:

Sano, M. et al., *Journal of the Physical Society of Japan*, 1990, 59(10), 3712-3719

Sano, M. et al., *Physical Review B*, 1989, 39(13), 9753

Based on these considerations, we can conclude firmly that 1L NiPS₃ has finite magnetic anisotropy that shows hexagonal symmetry.

In response to the referee comment, we added a statement in the revised version of manuscript. Also, Figure S9 is added to the revised version of supplementary information.

Figure S9. MR data on 1L device measured with an in-plane field perpendicular to edge #1 (green), compared to the data from **Figure 3c**.

Comment 2:

2. The data in Fig. 3C were collected at 40 K, where the signal-to-noise ratio is poor. Why were these measurements not performed at lower temperatures (e.g., 5 K, as in Fig. 4a), where noise might be reduced and the spin-flop signal is stronger?

Reply 2:

The choice of $T=40\text{K}$ is a compromise between several factors, as we explain below:

1. As shown in **Figure R3**, conductance of multi-layer crystals exhibits sudden decrease at low temperature (below approximately $T=25\text{K}$, grey-shaded region) independent of thickness. We think that this likely originates from the contact effect between few-layer graphite and semiconductor (NiPS_3) as similar trend of G vs. T has been observed in other 2D magnetic semiconductors, e.g. CrPS_4 [Wu, F. et al., *Advanced Materials*, 2023, 35.30: 2211653].
2. We therefore chose higher temperature of 40 K to avoid this effect across samples of multiple thickness and to ensure consistent comparison between different FET devices.
3. Lastly, $T=40\text{K}$ is significantly lower than T_N and therefore represents the low-temperature magnetic ordering of NiPS_3 . Our MR measurements on multi-layer crystals at $T=40\text{K}$ suggest that they exhibit the same low-temperature magnetic order and spin-flop field as bulk. This is also confirmed by low-temperature photoluminescence spectroscopy.

Figure R3. G as a function of temperature for 6L (red) and 2L (black) NiPS₃ devices. The grey-shaded region highlights the temperature range below 25K where conductance decreases rapidly. The conductance decrease at this temperature range is consistently observed independent of thickness across multiple devices.

Comment 3:

A local peak in the dG/dT vs. T curve marks the phase transition near 120 K. Since a second transition occurs around 60 K in 1L NiPS₃, is there any corresponding feature in the dG/dT vs. T curve?

Reply 3:

Thank you for pointing this out. **Figure S3** shows the G vs T and dG/dT vs. T curves of 1L NiPS₃ at different gate voltages, including the one measured at $V_{SD}=2V$, $V_{BG}=100V$ which corresponds to the same bias configuration used to construct the phase diagram of 1L by MR measurement (**Figure 4a, b**). Around $T=60K$, we do not observe distinct features in dG/dT vs. T curve.

The absence of a distinct dG/dT feature for the 2nd transition can be related to the fact that the conduction electrons experience similar scattering from the local spin configuration above T^* (quasi-long-range order) and below T^* (long-range order). On the other hand, the situation is different for a transition at T_N , which involves a much more significant change in local spin ordering—from paramagnetic (above T_N) to quasi-long-range order (below T_N), which can lead to more noticeable effect on electron conduction.

While preparing our reply, we realized that **Figure 4c** in the previous version of the manuscript compared MR data with the dG/dT vs. T data measured at different gate voltages (MR at $V_{BG}=100V$ and dG/dT vs. T at $V_{BG}=80V$). As shown in **Figure S3**, dG/dT vs. T curve measured at $V_{BG}=80V$ crosses zero at $T = 60$ K, corresponding to the minimum of G vs T curve. However, this zero-crossing in dG/dT should not be interpreted as an indication of second phase transition, as the appearance and the position of $dG/dT=0$ in 1L NiPS₃ is strongly gate-dependent. We note that the same gate-dependence is also observed in other thicknesses.

Figure S3. a-c. G (top panel) and dG/dT (bottom panel) as a function of temperature for 1L (a), 6L (b) and bulk ($> 20L$) (c) NiPS_3 , measured at different gate voltages. The position of peak in dG/dT does not show gate voltage dependence and corresponds to a magnetic phase transition (**Figure 1c, d**). Contrary, the temperature at which $dG/dT=0$ strongly depends on the applied gate voltage, and therefore should not be related to a magnetic phase transition.

In response to the referee comment, we added **Figure S3** to the supplementary information of the manuscript. Lastly, we updated **Figure 4b** to display the dG/dT vs. T curve measured at $V_{BG}=100V$, matching the bias configuration used for the MR data to ensure consistency.

Comment 4:

The spin-flop field in 1L NiPS3 (5 T) is significantly lower than in multilayer samples (10 T). Is this due to uniaxial anisotropy in thicker flakes? Could the authors provide theoretical calculations to explain this discrepancy?

Reply 4:

The lower spin-flop field in 1L NiPS_3 can be understood as a consequence of reduced magnetic anisotropy in 1L and a different anisotropy symmetry. In multilayer, uniaxial anisotropy is prominent, whereas in 1L, the data is consistent with predominantly hexagonal anisotropy.

According to modified spin-flop theory [H. W. Zandbergen, *J. Solid State Chem.* 35, 367 (1980), Sano, M. et al., 1989 *Phys. Rev. B* **39**, 9753(R)], a spin-flop transition can also occur in systems with hexagonal magnetic anisotropy, with a critical field given by $H_{sf} =$

$$\sqrt{\frac{6K_h}{\Delta\chi}}$$

where K_h is the anisotropy energy constant for hexagonal anisotropy and $\Delta\chi=\chi_{\perp}-\chi_{\parallel}$ is the susceptibility difference. For uniaxial magnetic anisotropy, the corresponding expression is $H_{sf} = \sqrt{\frac{2K_u}{\Delta\chi}}$ where K_u is the uniaxial anisotropy energy constant.

Following these relations, the observed difference in spin-flop fields between 1L and multilayers could plausibly be attributed to a larger uniaxial anisotropy constant K_u in multi-layers compared to hexagonal anisotropy constant K_h in 1L. Assuming that $\Delta\chi$ remains unchanged

across all thicknesses, a comparison of the spin-flop fields suggests that the anisotropy energy constant in 1L is approximately one order of magnitude smaller than that in the multi-layers. Detailed theoretical calculations tracing K and $\Delta\chi$ as a function of thickness would be valuable for a better estimation but lies beyond the scope of the present work. Nonetheless, our results are consistent with a suppression of the uniaxial component in the absence of interlayer coupling, leaving a weaker hexagonal term in 1L.

In response to the referee comment, we added a sentence in the manuscript highlighting the possible reduction of magnetic anisotropy energy in 1L.

Comment 5:

5. In Fig. S4b, the spin-flop field remains nearly constant below 50 K before increasing. What causes this apparent threshold behavior?

Reply 5:

The mean-field approximation of spin-flop fields provides a good explanation for our experimental observations. This is supported by the excellent proportionality between the extracted spin-flop fields (**Fig. S4b**) and the inverse square root of in-plane magnetic susceptibility anisotropy of bulk NiPS₃, as shown in **Figure 2e**. We believe that the nearly constant spin-flop field found below T=50K can be related to the rate of susceptibility anisotropy change at low temperatures, which tends to decrease — particularly below T =50K. At the same time, the accuracy of extracting spin-flop fields from the dMR/dB vs B data may not be sufficient below T= 50K for precise determination of their values.

Response to reviewers' critique for the manuscript “Nature of 2D XY antiferromagnetism in a van der Waals monolayer” by Cheol-Yeon Cheon, Volodymyr Multian, Kenji Watanabe, Takashi Taniguchi, Alberto F. Morpurgo, and Dmitry Lebedev.

Reply to Reviewer #4:

1. In this manuscript, the authors have investigated electrical transport behavior of NiPS₃ FET devices with thickness down to monolayer. Both the temperature dependence of conductance and MR results show a significant difference between the monolayer devices and the multi-layer(bulk) devices. Since previous researches applying optical approach can only detect long-range order in NiPS₃ down to two layers, the results of monolayer device are quite impressive. They discover that monolayer NiPS₃ undergoes two magnetic transitions, with a low-temperature phase governed by in-plane hexagonal magnetic anisotropy in contrast with the easy-axis zigzag antiferromagnetic order of the bulk NiPS₃. Generally speaking, this experimental work is well performed and the analysis of their data is rigorous, matching the level and interest of Nature Communications.

Reply 1

We would like to thank the reviewer for the positive assessment of our work.

Before I recommend publish of this manuscript, several of my minor points need to be addressed by the authors:

2. The authors have studied the layer-dependence properties of NiPS₃ with electrical methods. Recent studies have investigated the dimensional-crossover behavior in NiPS₃ with optical approach [Z. Sun et al, Nature Physics 20, 764–1771 (2024)] and non-local magnon spin transport method [B. Luo et al, SCIENCE CHINA Physics, Mechanics & Astronomy, 68, 127511 (2025)]. I believe these two articles are highly relevant to the current manuscript. I suggest the author add the discussion about the relationship between the manuscript and these two papers mentioned above.

Reply 2

Following the reviewer's suggestion, we added a discussion about recently proposed nematicity in few-layer NiPS₃ in our manuscript.

3. The actual order of NiPS₃ is controversial. The bulk NiPS₃ exhibits zigzag antiferromagnetic order with easy axis along the a-axis while below some critical thickness, the NiPS₃ exhibits a three-stage nematic vestigial order[Z. Sun et al, Nature Physics 20, 764–1771 (2024), B. Luo et al, SCIENCE CHINA Physics, Mechanics & Astronomy, 68, 127511 (2025)], which seems that can also fit with the multi-direction MR results in Fig.3 for the monolayer device. Is the low-temperature phase for monolayer governed by in-plane hexagonal magnetic anisotropy the same as three-stage nematic vestigial order? If yes, the author can use a uniform express as the references, if not, a discussion to distinguish the two phase is needed.

Reply 3

Recent reports brought by the reviewer propose that below thickness of 10 nm NiPS₃ does not transition into a conventional long-range order AFM state and instead evolves into to a 2D vestigial nematic state, where the long-range spin coherence is lost. In contrast to this conclusion, our magnetoresistance data for 2L, 6L and 13L NiPS₃ provide no indication that the nature of AFM order in bilayers and thicker multilayers deviates from that of bulk crystals. A final assessment of which state better explains the existing experimental data – a bulk-like long-ranged zig-zag AFM order or the newly proposed vestigial nematic phase – will require the investigation of how the vestigial nematic phase responds to an external magnetic field, which is currently unknown.

Reviewer is specifically asking about low-temperature phase of monolayer 1L NiPS₃. At low temperature, we observe field-induced spin-flop transitions and therefore we assign it to long-range AFM phase. As discussed in the manuscript, this assignment is consistent with renormalization-group analysis of anisotropic 2D XY model, which predicts long-range magnetic ordered phase at low-temperature.

Following the reviewer's suggestion, we added these considerations in the discussion.

4. The statement in line 123 “Application of external magnetic field along this axis results in a sudden 90° rotation of a Néel vector at a critical field value B_{sf} – a spin-flop metamagnetic transition.” is true for many commonly-seen easy-axis antiferromagnet but not accurate for NiPS₃. There are experimental signatures that for the easy-axis zigzag antiferromagnetic order in bulk NiPS₃, a spin-flop transition tends to pin the Néel vector along the a-axis, irrelevant of the direction of the magnetic field. [X. Wang et al, Nature Communications 15, 8011 (2024), B. Luo et al, SCIENCE CHINA Physics, Mechanics & Astronomy, 68, 127511 (2025)]. The statement can be changed into “Application of external magnetic field results in a reorientation of Néel vector at a critical field value B_{sf} – a spin-flop metamagnetic transition.” which is more universal and rigorous.

Reply 4

We modify our manuscript following the review's suggestion,

5. The monolayer NiPS₃ structure exhibits $D3d$ symmetry, while bulk NiPS₃ adopts a monoclinic stacking configuration with reduced $C2h$ symmetry. The layer-dependent magnetic anisotropy is relevant with the crystal symmetry. More discussion about this is also welcome.

Reply 5

Following the reviewer's suggestion, we included the symmetry of bulk and monolayer in discussion of magnetic anisotropy of NiPS₃.

6 The main concern of Reviewer 1 is how to distinguish the properties between the monolayer and multilayer samples. As claimed by the authors, one unique behavior of monolayer is the presence of two critical temperature: one is denoted by the peak of dG/dT as a function of T , another is evidenced by the emergency of MR and spin-flop transition as well as a fluctuation behavior of resistance at $T = 60$ K (Figure 4). Reviewer 1 suspects whether the bilayer and six-layer results are merely quantitatively, rather than qualitatively, distinct from the monolayer. This concern arises from the observation that in multilayer samples (e.g., the six-layer sample), there also exists a temperature range below T_C (~100 to 150 K) where no spin-flop transition is detected (Figure 2). In their response, the authors contend that the scenarios are fundamentally different. For the six-layer sample, the spin-flop field (H_{sf}) increases with temperature; at 100 K, it exceeds the maximum available magnetic field of 12 T. Thus, at higher temperature the transition is presumed to occur at fields beyond the experimental apparatus's range. In contrast, the monolayer's absence of a transition at its higher critical temperature is intrinsic and not measurement limited (Figure R1). I find the current data sufficient to support the manuscript's main conclusions and believe the requested comprehensive 2D scans are excessive.

Reply 6

We thank the reviewer for their concluding remarks and for the positive assessment of our data.

7 However, to directly address the Reviewer 1's valid point, I propose a more targeted experiment: extending the temperature-magnetic field (T - B) map for the bilayer device (Figure S8.b) to a higher temperature. The range should be extended to cover the bilayer's T_C (e.g., from $T = 10$ to 140 K and $B = -12$ to 12 T). If the H_{sf} in the bilayer also approaches 12 T before becoming undetectable, it would powerfully corroborate the authors' interpretation and clearly differentiate the multilayer mechanisms from the monolayer's unique characteristics.

In the meantime, Reviewer2's comments are well addressed by the authors.

Reply 7

Regarding the data for the 2L NiPS₃, we already provide multiple evidences which allow us to clearly differentiate its behavior from that of 1L NiPS₃. This includes:

- i) uniaxial magnetic anisotropy for 2L (matching 6L and bulk) vs hexagonal anisotropy for 1L;
- ii) critical spin-flop field around 10 T (matching 6L and bulk values) vs 5 T for 1L;
- iii) dependence of the spin flop field on temperature. We illustrate this point for 2L and 6L NiPS₃ on Figure 2e and Supplementary Figure S8b-c. For 2L, we observe a change of spin-flop field from 9.6 T at 20 K to 10.8 T at 90 K.